# Embryo-scale epithelial buckling forms a propagating furrow that initiates gastrulation

Julien Fierling[1,5], Alphy John [2,5], Barthélémy Delorme[2,5], Alexandre Torzynski[1], Guy B. Blanchard [3], Claire M. Lye[3], Anna Popkova[2], Grégoire Malandain [4], Bénédicte Sanson [3], Jocelyn Étienne [1,5✉], Philippe Marmottant [1,5], Catherine Quilliet[1,5] & Matteo Rauzi [2,5✉]

Cell apical constriction driven by actomyosin contraction forces is a conserved mechanism during tissue folding in embryo development. While much is now understood of the molecular mechanism responsible for apical constriction and of the tissue-scale integration of the ensuing in-plane deformations, it is still not clear if apical actomyosin contraction forces are necessary or sufficient per se to drive tissue folding. To tackle this question, we use the *Drosophila* embryo model system that forms a furrow on the ventral side, initiating mesoderm internalization. Past computational models support the idea that cell apical contraction forces may not be sufficient and that active or passive cell apico-basal forces may be necessary to drive cell wedging leading to tissue furrowing. By using 3D computational modelling and *in toto* embryo image analysis and manipulation, we now challenge this idea and show that embryo-scale force balance at the tissue surface, rather than cell-autonomous shape changes, is necessary and sufficient to drive a buckling of the epithelial surface forming a furrow which propagates and initiates embryo gastrulation.

---

[1] Univ. Grenoble Alpes, CNRS, LIPhy, 38000 Grenoble, France. [2] Université Côte d'Azur, CNRS, Inserm, iBV, Nice, France. [3] Department of Physiology, Development and Neuroscience, University of Cambridge, Cambridge, Great-Britain, England. [4] Université Côte d'Azur, Inria, CNRS, I3S, Nice, France. [5]These authors contributed equally: Julien Fierling, Alphy John, Barthélémy Delorme, Jocelyn Étienne, Philippe Marmottant, Catherine Quilliet, Matteo Rauzi. ✉email: Jocelyn.Etienne@univ-grenoble-alpes.fr; Matteo.Rauzi@univ-cotedazur.fr

The spectacular epithelial coordination that characterizes morphogenesis in the course of embryo development involves long range cell interaction. Previous studies support the idea that long-range tissue mechanics direct morphogenetic coordination[1–8]. Here we investigate the role of embryoscale mechanics in forming a furrow in an epithelial tissue. Furrow formation is a process that involves the bending of a tissue along a line. The formation of a furrow is pivotal during embryo development since it initiates tissue topology changes structuring the future animal. Unravelling the mechanisms which drive furrowing is thus key to understanding how vital processes such as gastrulation and neurulation are initiated. Ventral furrow formation (VFF) during early gastrulation in the *Drosophila* embryo is a well studied process: a fold along a line at the ventral side of the embryo (the prospective mesoderm) forms parallel to the anterior–posterior (AP) direction[9,10]. VFF is eventually followed by ventral tissue internalization and germband extension. In vivo studies have highlighted several concurring phenomena during VFF. Cell apical constriction[11], basal expansion[12], cell lengthening[13,14] and lateral tension[15–17] are phenomena that have been studied and that are associated with cell shape changes (e.g., from columnar to wedge-shaped) and the eventual formation of the furrow.

Cell apical constriction driven by the contraction of apical actomyosin networks is one of the most thoroughly investigated mechanisms, since it is a striking change in shape, given its magnitude and rapidity, and it is amenable to analyse as it occurs at the surface of the tissue where microscope-based imaging technologies can be more easily applied. Numerous studies have highlighted the molecular nature and the key role of apical constriction in driving furrow formation[11,18–21]. Coordination of apically constricting cells along an anisotropic area has also been shown to be responsible for stress anisotropy at the tissue surface[22] and to be associated with furrow formation[23]. However, the currently accepted mechanism driving VFF is based on cell apical-basal differential tension (or cell lateral tension) and cell autonomous wedging[24] and does not require or explain three-dimensional morphogenesis coordination.

Computational modelling has been used to tackle the biophysical mechanisms responsible for the prospective mesoderm shape changes resulting in VFF. 2D models, mimicking the embryo cross section, have predicted that apical constriction per se may not be sufficient to drive furrow formation and that basal and lateral forces may be necessary to initiate tissue furrowing[24,25]. A combination of apical and basolateral forces can locally invert the spontaneous curvature of the model epithelium and fold it[24–30]. 3D models have also been proposed and can produce an AP-oriented furrow with a more realistic 3D geometry, qualitatively comparable to in vivo morphology[31–34]. However, these models have similar biophysical working principles as 2D models and have not been extensively tested against specifically 3D features of VFF.

By implementing an experimental strategy combining computational modelling, infra-red femtosecond (IR fs) laser manipulation coupled to multi-view light sheet microscopy and quantitative image analysis, we unveil an emergent long-range mechanism powered by apical contraction forces and based solely on surface mechanics (i.e., free of cell lateral and basal forces and not relying on cell shape changes from columnar to wedged) over a curved tissue that is sufficient to drive the formation and the propagation of the ventral fold. With the aim of setting common grounds, we provide in Box 1 the definition of key terms that will be extensively used in this manuscript.

## Results

**Tissue-scale IR fs laser ablation of apical actomyosin prevents furrow formation**. A large body of genetic perturbation experiments,

resulting in the eventual failure of cell apical constriction and mesoderm invagination, support the idea that apical contraction of actomyosin networks is key to drive VFF[22,35–39]. Therefore, the in-plane forces at the surface of the embryo, generated by the apical contractile network[22], are necessary to drive a tissue-scale furrow. In order to test the role of actomyosin contraction forces in mesoderm invagination with spatio-temporal specificity, Guglielmi and colleagues in 2015 developed a two-photon optogenetic technique that allows to indirectly perturb the actin cytoskeleton[23,40–43]. This study shows that depletion of cortical PIP2 (a plasma membrane phosphoinositide) in mesoderm cells inhibits apical constriction and mesoderm invagination. We now aimed to perturb more directly the actomyosin network to specifically assess the role of apical contraction forces in mesoderm furrow formation. To that end, we took advantage of IR fs laser ablation to sever the actomyosin network with high spatio-temporal specificity without compromising cell membrane integrity (as shown in previous studies[44,45]). We now performed tissue-scale IR fs ablations of the actomyosin supracellular network across the dorso-ventral (DV) width of the mesoderm (Fig. 1a and Supplementary movie 1). After laser ablation, the network is cut and recoils while the cell apical surface membrane is preserved and dilates. The actomyosin network eventually recovers, restoring apical contraction forces and cell apical constriction (Fig. 1a and Supplementary movie 1). The recovery of both the actomyosin network and the shape of cells corroborates the notion that IR fs ablation is an effective tool to sever contractile actomyosin networks while preserving cell integrity and viability. Next, we performed ablation of the actomyosin network during furrow formation (i.e., when the ventral-most tissue adopts a concave shape, *t*1 in Fig. 1b) and monitored tissue curvature changes. After ablation, the ventral-most tissue loses its concave curvature, regaining its original convexity (*t*2 in Fig. 1b, c). Eventually, after actomyosin network recovery, the ventral furrow regains its distinctive concave shape (*t*3 in Fig. 1b, c). This directly demonstrates that apical contraction forces, driven by apical actomyosin networks, are necessary for VFF. After VFF, the mesoderm tissue is internalized. To test whether apical contraction forces are also necessary for subsequent tissue internalization, we performed repetitive laser ablations of the ventral actomyosin network in order to impede its recovery. Under these conditions the ventral cells remain at the embryo surface and tissue internalization fails (Supplementary movie 2). Overall, this reinforces the idea that apical contraction forces generated by actomyosin networks are necessary for both furrow formation and subsequent tissue internalization.

**An active surface model to infer sufficient conditions for furrow formation**. We next asked whether other forces than active apical contractility are necessary to form a furrow. During VFF, the *Drosophila* embryo is constituted of a single layer epithelium with the cell apices facing outwards (see Fig. 1d). Based on 2D modelling, it has been claimed that active basolateral forces are required for furrowing[25,32]. Optogenetic myosin II (MyoII) activation in the vicinity of the apical surface has shown that furrowing can be achieved by triggering active contractility[46,47]. However, passive basolateral forces could also be at play, resulting from the elastic modulus associated to cell lengthening[24] or to a constant cell length being imposed[33]. In embryos carrying mutations for the genes *slam* and *dunk* (*slam⁻dunk⁻*), the process of cellularization (prior to VFF) is impaired: the apical membrane fails to ingress to form lateral and basal sides of blastoderm cells, resulting in acellular embryos[14]. In acellular embryos, nuclei of mesoderm cells still displace basally[14] and a ventral furrow still forms (Fig. 1d). Therefore, we asked what mechanism could allow apical forces alone to lead to deformations similar to those observed in vivo. We modelled the apical surface of the *Drosophila* blastoderm as a thin elastic surface

**Box 1 ▌ Definition of key terms**

*Furrowing*: the process by which a tissue bends forming a fold having much greater bending curvature along one direction compared to the orthogonal direction. *Invagination*: the process by which a tissue is internalized inside the embryo. *Buckling*: a sudden change in the type of mechanical equilibrium, from a state in which load is mostly balanced by internal forces along the plane of the tissue to a state in which load is mostly balanced by internal forces normal to the tissue. *Strain*: the magnitude of deformation of material having a given shape (called the current configuration) at a specific time and location with respect to its equilibrium configuration. In the absence of pre-strain, the equilibrium configuration is the original shape. *Pre-strain*: strain associated with a change of the equilibrium configuration rather than changes of the current configuration. For instance, an increase of the activity of the molecular motor MyoII corresponds to an increase in pre-strain, not necessarily entailing a change of shape (current configuration) with respect to the original shape. *Stress*: the internal tension within the material which is elicited by strain. *Pre-stress*: stress which is due to a change of the equilibrium configuration (i.e., the pre-strain).

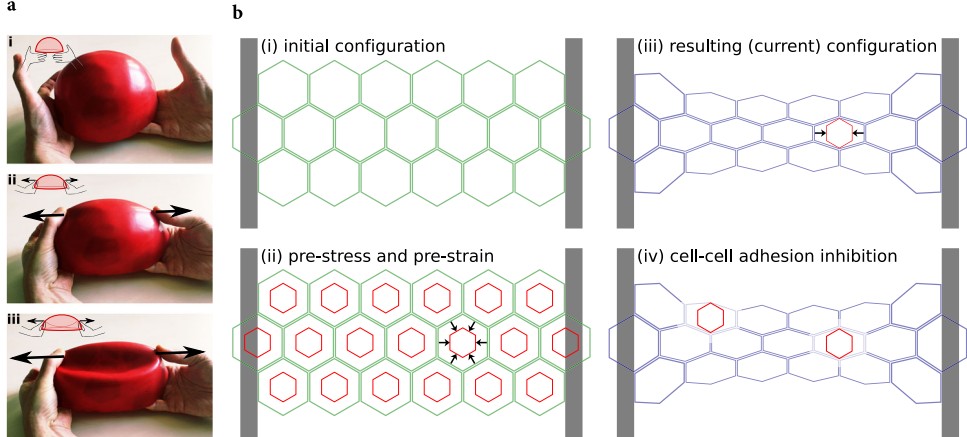

**Boxed Fig. 1**. (a) An example of buckling of an elastic sheet. (i) A thin elastic shell in its equilibrium configuration. (ii) Forces applied in two positions deform the elastic shell. Note the flattening (reduction of curvature) and the elongation (in-plane deformation). (iii) Above a threshold level of applied force, buckling occurs. Note the depression (inversion of the sign of curvature) surrounded by a highly curved rim. The change of shape happens suddenly. (b) Deformations and stresses in a tissue with anisotropic boundary conditions and isotropic actomyosin contractile activity. (i) Initial configuration of a tissue held between two clamped boundaries at its longitudinal ends and free at its latitudinal ends. Without MyoII activity, this represents the equilibrium configuration. (ii) Representation of the effect of uniform isotropic MyoII activity. The initial configuration (green) is no longer the equilibrium configuration: the red shape shows the equilibrium configuration of each cell, which is thus pre-strained in its initial configuration. Black arrows indicate the corresponding pre-stress (here shown for one cell). (iii) Current configuration (blue) resulting from MyoII pre-stress. Tissue cohesion and fixed boundary conditions prevent cells from adopting their equilibrium configuration (shown in red for one cell). Along the latitudinal direction, away from the boundaries, cells are able to contract to their equilibrium size, thus strain (nearly) equals pre-strain in this direction, which results in (close to) zero stress along this direction. Along the longitudinal direction, tissue cohesion and boundary conditions prevent any length change compared to the initial shape, thus strain is zero and stress (arrows) is equal to pre-stress along this direction. (iv) If cell-cell cohesion is inhibited, cells relax to their equilibrium shape (red), revealing the anisotropy of stress in the configuration (iii)[22,56].

(Fig. 1e). A more general approach would have been to consider a visco-elastic model, since the relaxation time of the *Drosophila* embryonic epithelium is estimated to be around one minute[48], shorter than the process of VFF[4]. However, a visco-elastic model would have involved both a much higher computational complexity and a much greater difficulty of exploring the parameter space to fit the data. Since the mechanical load due to molecular motor MyoII activity is constantly increasing during the early phase of the VFF process[33], we hypothesized that the effects of the viscous relaxation would be negligible in comparison to the elastic response to the increased load (see Supplementary Information). Under this hypothesis, a purely elastic model is sufficient to recapitulate the main features of this fast process.

We modelled the initial three-dimensional shape of the *Drosophila* embryo as in ref. [5] (Supplementary Information). The global shape of the embryo is ovoid, with the long axis (along AP) three times the length of the short axis (along DV). In good approximation to live embryos, our model presents a dissymmetry with respect to the mid-coronal plane, the ventral side being more curved than the dorsal, while it is symmetric with respect to the mid-transverse and mid-sagittal planes. As in previous models, the volume within this elastic surface is assumed

to be constant throughout the simulations. The embryo shape is also constrained by the vitelline membrane, which is undeformable, and separated from it by a thin layer of perivitelline fluid. We assumed that, before VFF, the distance between cell apices and the vitelline membrane is approximately 0.2−0.5 μm. This distance is allowed to vary locally[27,49–51] while the global volume of perivitelline fluid is kept constant. We assume that tangential forces, such as friction or specific adhesion between the apical elastic surface and the vitelline membrane, are negligible at this stage, consistent with experimental and theoretical results[5,50,51]. We also hypothesize that viscous shear forces exerted on the surface are negligible.

Using the finite element software *Surface Evolver*[52], we model the mechanics of the apical surface of the embryo as a thin elastic shell of given elastic properties[53]. In this continuous model, the tissue is not partitioned into cells, nevertheless regions of space can still be assigned different properties, reflecting distinct portions of the embryo showing different gene expression patterns and mechanical properties[5,54]. Therefore, in order to represent local MyoII activity (as in previous studies[5,55]), we assign a local active pre-stress to finite element facets in the region corresponding to the embryo prospective mesoderm (see

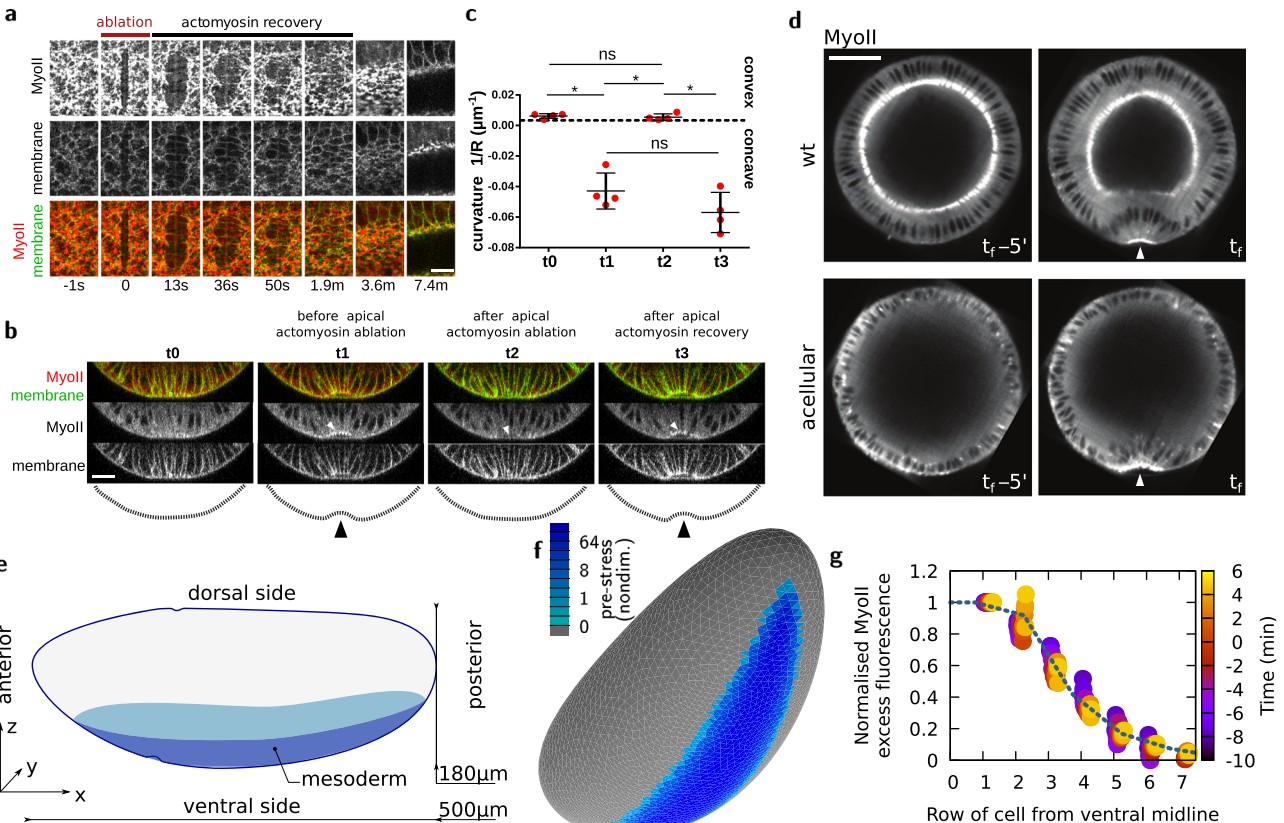

**Fig. 1 Using IR fs laser ablation to demonstrate the necessity and building of a model to test the sufficiency of apical contraction to drive VFF. a** Recoil and recovery of the ventral apical actomyosin network after laser ablation. The ablation is performed along the DV axis across the ventral tissue. MyoII is depleted along the ablated region while membrane signal density eventually decreases as a consequence of cell dilation. Panel shows representative experiment. Experiment repeated 5 times on 5 different embryos. Scale bar 10 μm. **b** Cross-sectional view of the embryo ventral side just before (t0 and t1) and just after (t2) laser ablation, and (t3) after actomyosin recovery. Time between t0 and t3 is 3 min. Laser ablations were performed similarly as in (**a**). Panel shows representative experiment. Experiment repeated 4 times on 4 different embryos. Scale bar 20 μm. **c** Curvature analysis before and after laser ablation and during actomyosin network recovery as shown in (**b**). $n = 4$ embryos, data are presented as mean values ± standard deviation. The statistical test performed was Kruskal–Wallis test for multiple comparisons, $*p \leq 0.05$ and ns (non-significant), $p > 0.05$. **d** Digital mid-cross-sections before and during furrow formation in wild-type and $slam^- dunk^-$ acellular embryos. Panel shows representative case, $n = 4$ embryos. Scale bar 50 μm. **e** Representation of the embryo geometry with the mesoderm region highlighted. **f** Finite element mesh of an embryo-shaped elastic surface where some facets will be pre-strained to mimic MyoII activity (colour code, log scale, nondim. nondimensional units). **g** Circles, MyoII profile at different phases of VFF as a function of distance from the ventral midline, normalized using the intensity of cells at midline (row 0, see the "Methods" section), in experiments (average of $n = 3$ embryos, confocal microscopy). Line, pre-stress profile chosen for simulations. This profile is also similar to the one reported by ref. [33].

Fig. 1f). We do so by assigning these facets a target area $A_0$ which is smaller than their initial area $A_i$ (Fig. 2a). This means that, before deformations cause relaxation, these facets experience pre-strain $\varepsilon_a = \frac{A_i - A_0}{A_0}$ and thus are pre-stressed by $\sigma_a = \chi_{2D} \frac{A_i - A_0}{A_0}$, where $\chi_{2D}$ is the in-plane compression elastic modulus (see Supplementary Information and Supplementary Fig. 1b). We find that the pre-stress applied in the mesoderm region results in increased tension not only at the ventral but also at the lateral and dorsal regions of the embryo (Fig. 2b). For an explanatory definition of pre-strain and pre-stress (see Box 1).

It has been observed that MyoII activity is not uniform across the mesodermal cells: MyoII shows a graded distribution over approximately seven AP rows of cells on both sides of the mesoderm with highest level at the mid-line[33]. Experimentally, we have found that this spatial profile is well preserved during VFF (see Fig. 1g and the "Methods" section). We have thus implemented a graded distribution of pre-stress in our computational model mimicking the graded MyoII distribution. To this end, we have applied a gradient of pre-strain which starts from the ventral mid-line where $\varepsilon_a$ is equal to $\varepsilon_a^m$ and decreases according to the profile shown in Fig. 1g. The resulting stress is largest in the region where the pre-stress is applied, but decreases laterally with a profile that differs from the one of the pre-stress gradient (Fig. 2b).

In sum, in order to test if embryo surface mechanics are sufficient to drive VFF, we have developed a computational model devoid of cells based on a single thin elastic sheet that closely reproduces embryo 3D shape and the mechanical properties at the blastoderm surface. This is different from previous computational models[24–33] that rely also on active or passive cell basal-lateral forces.

**Tension anisotropy emerges from tissue and embryo geometry.** We next asked whether this model is able to reproduce known tissue-scale mechanical properties of the *Drosophila* prospective mesoderm. The increase in tension in the mesoderm during VFF is known to be anisotropic, with a higher tension along the AP axis than along the DV axis[22]. It has been shown that tension anisotropy is the cause of the anisotropic organization of the actomyosin network that eventually arises during VFF[56]. It has been suggested that the geometry of the embryo[56,57] and of the contractile pattern[23,46] are responsible for the emergence of this

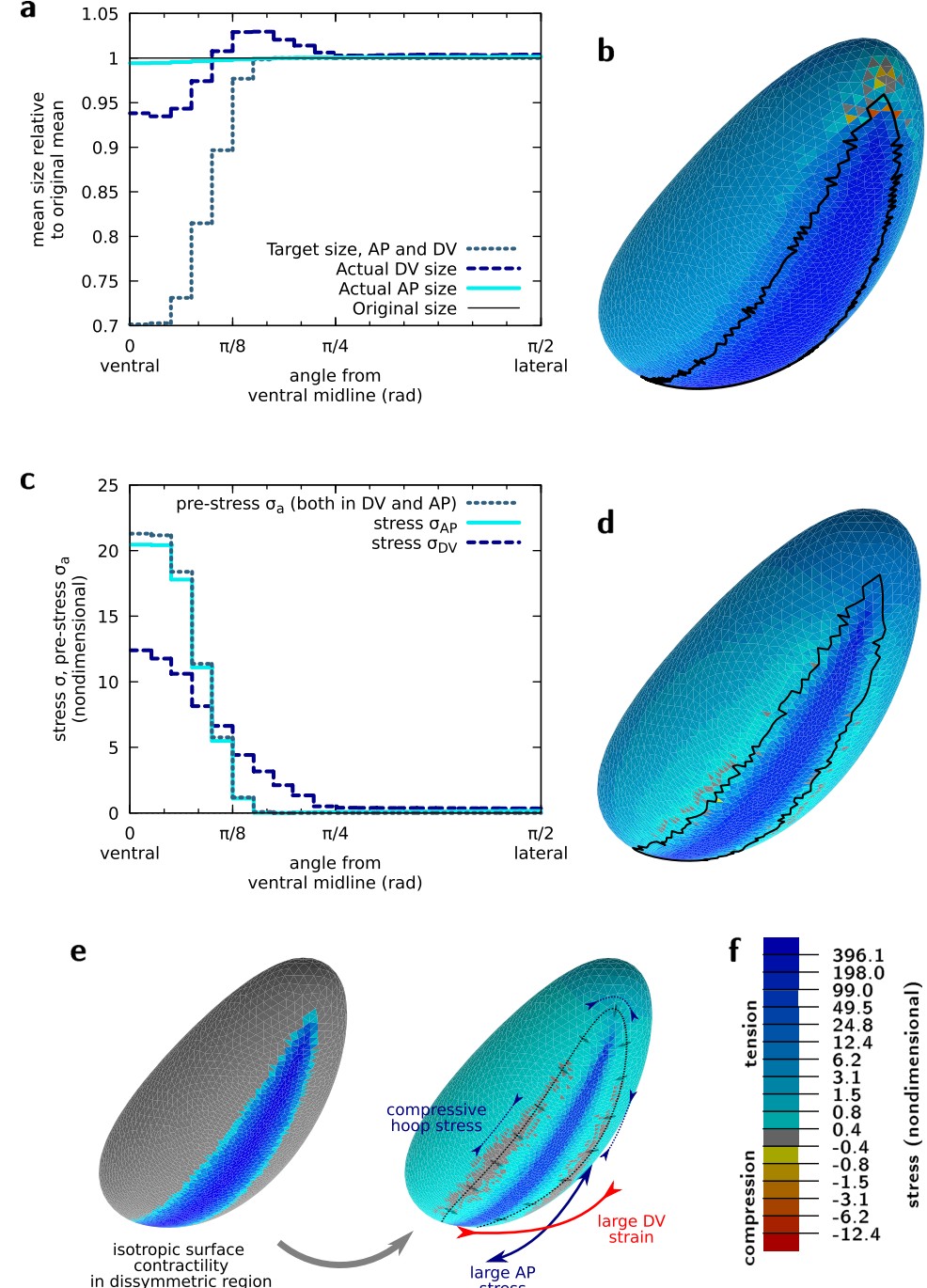

**Fig. 2 Actomyosin contractility drives tension anisotropy on the ventral side of the embryo. a** Strain angular profile (current size relative to initial size) along the AP and DV axes for midline pre-strain $\varepsilon_a^m = 0.43$. **b** Mechanical stress (sum of the two principal stresses) resulting from the area pre-strain. Midline pre-strain $\varepsilon_a^m = 0.43$. Black line corresponds to the boundary of the pre-strained region in the current configuration. See tensor components in Supplementary Fig. 1b. **c** Angular profiles of the pre-stress $\sigma_a$ and of the two principal stresses along the AP and DV axes for $\varepsilon_a^m = 0.43$. **d** Same as (**b**) for $\varepsilon_a^m = 5.25$. See tensor components in Supplementary Fig. 1d and profiles in Supplementary Fig. 1e, f. **e** Isotropic pre-strain pattern (left) yields anisotropic mechanical response, with a greater stress and strain along the AP and DV axes, respectively. The cells at the periphery of the mesoderm move towards it, arrows, which generates a hoop stress along the dotted line. **f** Colour code for panels (**b**) and (**d**). All panels are for nondimensional mechanical parameters $\tilde{\chi}_{2D} = 50$ and $\nu_{2D} = 0$, see Supplementary Information.

anisotropy of tension. We thus considered whether a mechanical model would predict anisotropic tension by imposing isotropic MyoII activity. In an elastic flat plate, it has been shown that isotropic active pre-stress in an asymmetric geometry causes stress and strain anisotropy, which are, respectively, larger along the long and short axis of the domain[58,59]. Our simulations show that this effect is also at play in the embryo 3D geometry, and that

mechanical tension is strongly anisotropic in the mesoderm, with AP tension twice the DV tension along the ventral midline (Fig. 2c, e and Supplementary Fig 1a, b).

What is the origin of the stress anisotropy? To answer this question, we compared the dimension of the ventral tissue with respect to the dimensions of the entire blastoderm along the AP and DV axes. The ventral tissue is about three and six times less

than the total blastoderm length along the mid-sagittal and mid-cross sections, respectively (Supplementary Fig. 1c). Therefore, the intensity of the stretch of neighbouring tissues which is necessary to achieve the same contraction strain along the two orthogonal directions is about three times greater along the AP than the DV axis. This results in a three times greater cell resistance to stretch along AP than DV. The shape anisotropy of the system would thus explain why the surrounding tissue appears more difficult to deform along the AP than along the DV direction even though mechanical properties of the entire tissue surrounding the mesoderm are imposed to be the same.

The AP stress $\sigma_{AP}$ is found approximately equal to the pre-stress $\sigma_a$ initially imposed on the ventral region. This is consistent with the fact that little tissue deformation takes place along the AP axis (Fig. 2c), and can be related to the mechanical behaviour of contracting actomyosin networks anchored to stiff boundaries which resist deformation[60]. On the other hand, the DV stress $\sigma_{DV}$ is found to be different from the pre-stress, both in the ventral and in the ventro-lateral region. This is consistent with the differential tissue deformation taking place along the DV direction: in the highly contractile ventral region (for angles between 0 and $\pi/8$) the tissue contracts (Fig. 2a) resulting in $\sigma_{DV}$ smaller than $\sigma_a$ (Fig. 2c). In the ventro-lateral region (for angles between $\pi/8$ and $\pi/4$) the tissue passively stretches resulting in $\sigma_{DV}$ greater than the local value of $\sigma_a$. If pre-stress is increased by a multiplicative factor, mimicking the increase of MyoII activity observed in vivo, these features are dramatically accentuated (Fig. 2d–f, Supplementary Fig. 1a, d–f).

The trace of the stress tensor, presented in Fig. 2b, d, is positive everywhere except at the anterior and posterior ends of the mesoderm region which are under net compressive stress in Fig. 2. However, the principal stresses, which are the eigenvalues of the stress tensor, show a more complex pattern (Supplementary Fig. 1b, d). While both principal stresses are positive in the mesoderm, one of them is negative, denoting directional compression, just beyond the periphery of the mesoderm. The compressive stress is oriented orthogonally to directions pointing to the ventral furrow: it is thus parallel to the AP axis at lateral positions and parallel to the DV axis at anterior and posterior positions. This feature was not reported in 2D planar models[58,59]. This compressive pattern can be intuitively understood by the fact that cells beyond the periphery of the mesoderm move centripetally towards the ventral midline, which generates a compressive hoop stress (Fig. 2e).

In sum, we show that our model reproduces tension anisotropy on the ventral side of the in silico embryo as an outcome of isotropic MyoII pre-stress, highlighting the establishment of tensile and compressive stress patterns that result intrinsically from geometry.

**In vivo surface shape changes are reproduced by the mechanical model.** In order to validate our mechanical model, we compared the dynamics of area change obtained computationally to experimentally measured changes in cell surface area while imposing a localized pre-stress increase proportional to MyoII intensity distribution measured experimentally in both space and time (Figs. 1g and 3a). MyoII activity increases strongly in the mesoderm as VFF engages. Quantitatively, the evolution of background-subtracted MyoII fluorescence in cells at the ventral midline is well approximated with a double-exponential function:

$$I_{myo}(t) = I_c + I\left(e^{e^{(t-t_0)/T}-1} - 1\right) \qquad (1)$$

where $I_c$ and $I$ are coefficients in arbitrary units of fluorescence intensity, $t_0$ the time from which MyoII shows a progressive increase, and $T$ the characteristic time of increase of MyoII. The

rate of these dynamics, reflected in the constant $T$, depends on temperature, however the time profile is highly reproducible for different experiments (Fig. 3a, see the "Methods" section). We fitted the pre-stress with MyoII temporal dynamics for an optimal match with the first part of the process of furrow formation (i.e., until minute 1 in Fig. 3a). In our computational model, as pre-stress gradually increases, the ventral region reduces in area showing similar dynamics as measured in vivo (Fig. 3a). In vivo, cells located close to the ventral midline exhibit an area reduction which is proportional to MyoII intensity increase. By contrast, cells located further away from the midline (between the 6th and the 9th cell row) increase in area (Fig. 3b) although their MyoII intensity also increases at a lesser rate (Fig. 1g). This corroborates previous experimental evidence and quantitative analysis showing that cells further away from the midline increase their surface area during furrow formation[4,10,33,58,61,62]. This spatial strain pattern is reproduced by our computational model (Fig. 3b). These results give rise to a seeming paradox: how can cells increase their surface area while simultaneously increasing the level of MyoII (i.e., the constricting pre-stress $\sigma_a$)? To find an answer to this question we further investigated cell shape changes by quantifying the AP and DV lengths of cells located at different DV positions from the ventral midline. Cells with increasing surface area show a significant increase in DV length (Fig. 3c, d, Supplementary Fig. 2a) in agreement with anisotropic tissue strain shown in Fig. 2a. Thus, we considered whether this area increase was directly linked to the DV extension found in simulations in ventro-lateral locations. Figure 3c–f shows indeed that all cells more than three-cell radii from the midline are generally stretched along the DV direction from an early time in gastrulation, in fair quantitative agreement with simulations (see also Supplementary Fig. 2a and Supplementary movie 3). Consistent with our model and in agreement with Bhide, S. et al.[61], cells for which neighbouring cells exert an extrinsic tensile stress larger than their own MyoII-based pre-stress are thus being stretched towards these neighbouring cells. This effect could be additionally enhanced by local modulations of the mechanical properties of filamentous actin within the mesoderm[62], resulting in an even larger stretch in vivo than predicted by the model. The contraction along the AP direction, which is observed experimentally and which simulations predict (Supplementary Fig. 2a, c–e), is of smaller magnitude for cells farther from the midline. Therefore the combination of DV stretch and AP contraction results in an area increase for cells located on the ventro-lateral position. These results show that our model can faithfully reproduce the in-plane tissue deformation observed in vivo.

**Apical actomyosin mechanics drive AP midline flattening and furrow propagation.** Our model is now (i) tuned by imposing a pre-stress that is proportional to the experimentally measured MyoII distribution and (ii) validated since the imposed pre-stress results in surface area changes of the elastic sheet in good agreement with cell apical surface changes measured in vivo. Remarkably, simultaneous to these surface area changes, the 2D elastic sheet forms a buckle resulting in a furrow along the long axis of the 3D ellipsoid in the region under pre-stress (Fig. 4a and Supplementary Fig. 3a). This shows that forces applied at the surface of an ellipsoidal 3D shape can be sufficient to drive the formation of a furrow. This implies that in vivo, such a 3D mechanism could allow apical contractility to form the ventral furrow[30], without the assistance of active or passive basolateral forces.

We then tested the predictive power of our model. We analysed the furrow at different positions from the poles: the furrow forms first and is deeper in the mid region of the ellipsoid and appears

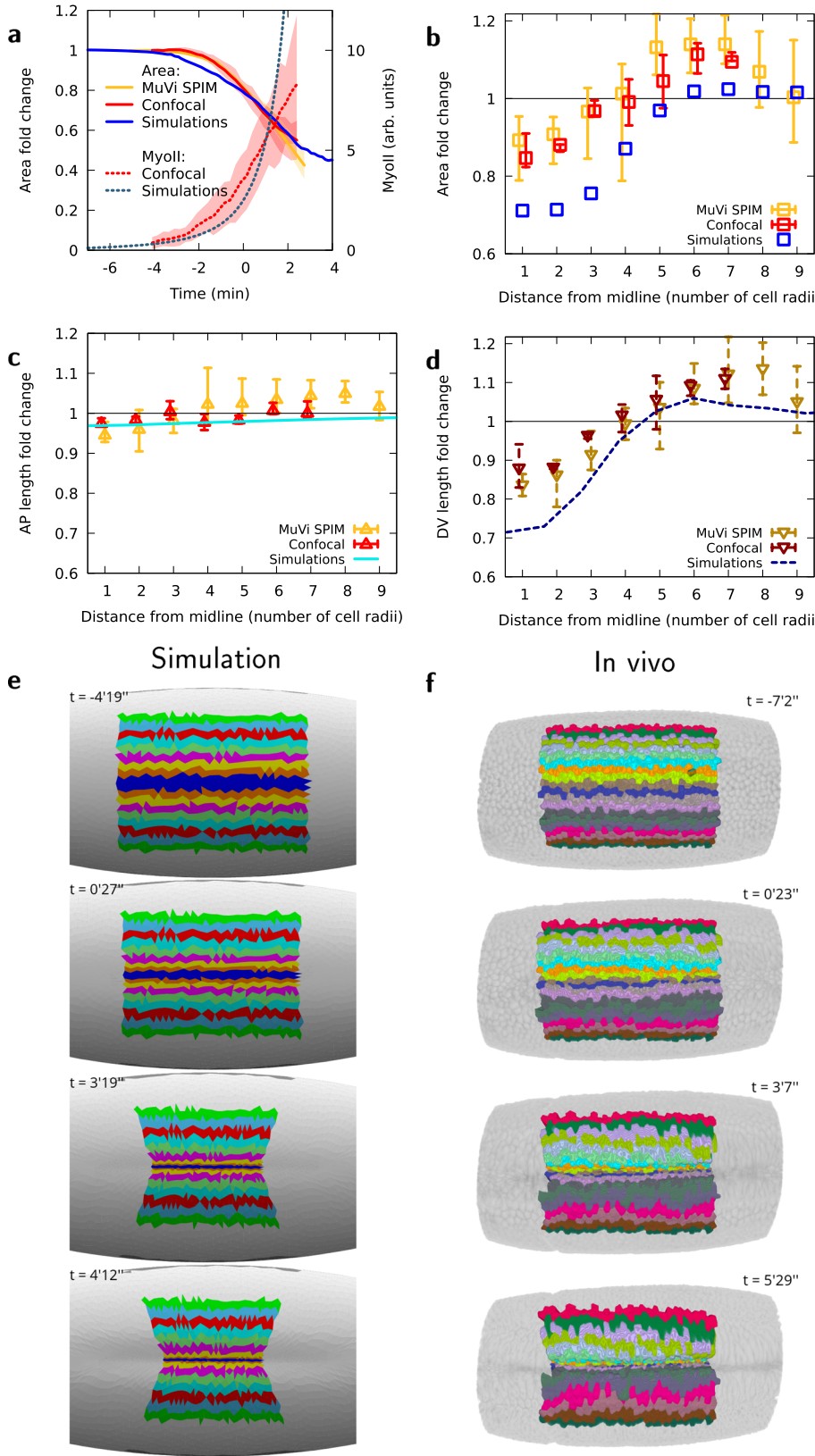

with some delay in regions closer to the poles (Fig. 4b, c). We then measured furrow depth at different AP positions in vivo. To that end, we embedded the embryo in a soft gel cylinder (preserving the embryo shape) and imaged it with multi-view light sheet microscopy (MuVi SPIM) to obtain isotropic resolved images (see the "Methods" section). The embryo shows the same features as the computational model: the furrow forms deeper in the mid-region of the embryo and eventually propagates towards the anterior and posterior poles (Fig. 4b–d, Supplementary movie 4), even though MyoII apical accumulation and cell apical constriction are uniform along the AP axis and do not form a propagating wave (Supplementary Fig. 3b, c). The curves

**Fig. 3 In vivo apical area changes are reproduced by the computational model. a** MyoII average intensity and mesoderm apical area changes as a function of time, for in vivo analysis and simulations, averaged over all cells within five rows of the ventral midline. **b** Apical area fold-change relative to the initial area ($t = -4$ min) of cells at different lateral distances from the midline at $t = -1$ min, in observations for in vivo analysis and simulations. (**c**) Apical AP size fold-change relative to the initial size ($t = -4$ min) of cells at different lateral distances from the midline at $t = -1$ min, **d** Apical DV size fold-change relative to the initial size ($t = -4$ min) of cells at different lateral distances from the midline at $t = -1$ min, for in vivo analysis and simulations. Panels (**a**–**d**), $n = 3$ embryos using multi-view light sheet and $n = 3$ embryos using confocal microscopy, shaded areas in (**a**) and error bars in (**b**–**d**), minimum and maximum values among the corresponding embryos. **e**, **f** Time evolution of AP stripes of apical surface in simulation and MuVi SPIM, respectively.

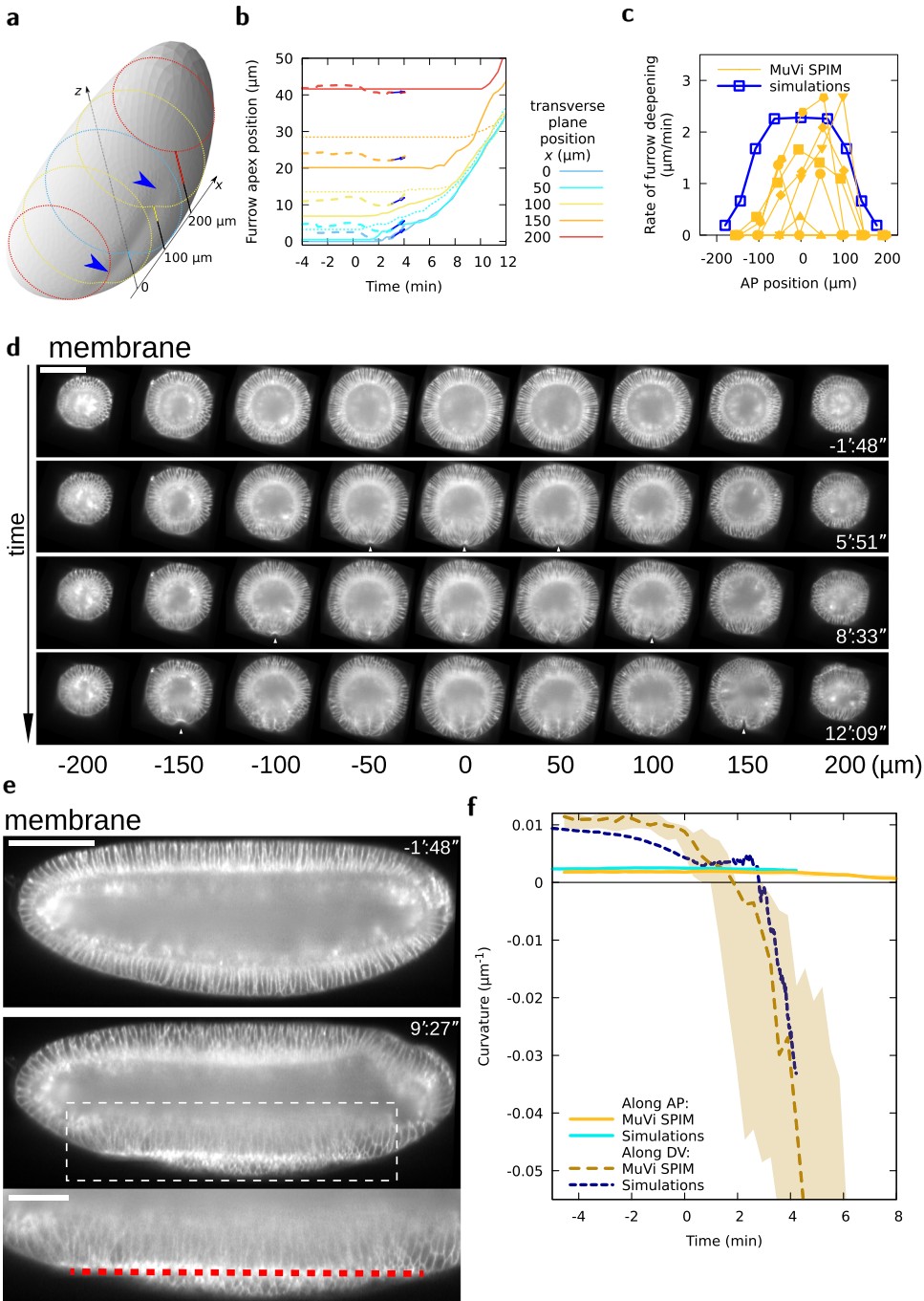

**Fig. 4 VFF results from tissue curvature changes along the DV and AP axes. a** Embryo shape during VFF in simulations at $t = 4'12''$. Shading reveals furrow shape, blue arrowheads. Dotted lines are transverse cuts, solid lines give the furrow apex offset from a reference $z$ position at different $x$ positions. **b** Furrow apex position at different AP positions as a function of time in MuVi SPIM experiments (solid lines, posterior side, dotted lines, anterior side, $n = 6$) and simulations (dashed lines and arrow showing slope at $t = 3'$). **c** Rate of furrow formation at different AP positions at $t = 3'$. **d** Digital cross-sections at different AP positions. White arrowheads indicate VFF initiation. Scale bar 100 μm. **e** Digital mid-sagittal section of the embryo. Red line indicates ventral tissue flattening. Scale bars 100 μm, zoom 50 μm. **f** Curvature of the ventral tissue along the AP and DV axes as a function of time (MuVi SPIM, $n = 6$ embryos, shaded area denotes minimum and maximum).

representing the absolute furrow apex position at different AP locations in the embryo over time, eventually merge, highlighting a remarkable feature of VFF: while folds at distinct AP positions initially form at a different rate, eventually they align, sequentially reaching the same absolute apex position (Fig. 4b, Supplementary Fig. 3d and Supplementary movie 5). To better decipher the dynamics of ventral furrow propagation in the embryo, we digitally sectioned the 3D image of the embryo along the mid-sagittal plane (the plane separating the left from the right side of the embryo and intersecting the furrow midline, Supplementary Information). The mid-sagittal view of the embryo reveals a new feature: during furrow formation, the ventral tissue midline flattens along the embryo AP axis (Fig. 4e and Supplementary movie 4). Interestingly, the acellular embryo shows similar furrow formation features (Supplementary movie 6). While ventral flattening along DV has been characterized previously[9], AP flattening along the ventral tissue midline can only be seen in a mid-sagittal view which has not been reported before. To better characterize the dynamics of furrow formation in 3D, we measured and analysed the changes of ventral tissue curvature along both the DV and AP tissue midlines (Fig. 4f, Supplementary Fig. 3e). Ventral curvature analysis for both our computational model and the embryo show the same trend: the curvature of the AP ventral midline gradually decreases eventually reaching the zero value (i.e., a flat tissue), concomitantly the ventral DV curvature decreases much faster until suddenly transitioning from positive to negative values (i.e., from convex to concave). Although this process is abrupt, it is not discontinuous (see Supplementary Fig. 3f) and it starts from a finite threshold value of pre-stress.

Overall, this shows that our mathematical model has unprecedented predictive power and reveals 3D features of VFF that were not reported in previous studies, such as mesoderm tissue AP flattening and furrow propagation.

**The polar caps function as anchoring sites for the contractile ventral tissue.** The ventral region of the ellipsoidal elastic sheet contracts under pre-stress (Fig. 2) resulting in sheet deformation. While along the DV axis the ventral sheet constricts in the mid region and stretches in more lateral regions, along the AP axis the sheet shows less deformation (Fig. 2a, Supplementary Fig. 1e, Fig. 3c–f). Since AP pre-stress drives little AP strain, it results in stress within the contractile region which can only be balanced by forces acting upon neighbouring tissues. We thus focused on the neighbouring tissue forming the polar caps. The polar tissues are submitted to both pressure forces exerted by the incompressible cytoplasm (Fig. 5b, grey arrowheads) and by pulling forces exerted by the contracting ventral sheet (Fig. 5b, red arrows, and d). During sheet contraction, the polar caps are pulled inwards leading to an increase of perivitelline space at the poles (Fig. 5a, b, dashed line). Remarkably, the same process occurs also in vivo (Fig. 5c and Supplementary Fig. 3g) and tension is also maximal close to the midline (Fig. 5e). Therefore, the anterior and posterior polar caps may function as symmetrically positioned anchoring sites between which the ventral tissue midline flattens, working as a contractile string driving furrow formation. These 'anchors' are not immobile and result from the combination of the embryo geometry, internal pressure, and epithelial cohesion that provide resistance to the polar cap displacement towards the ventral region. To test if the position of anchoring sites could bias AP tissue midline flattening, we implemented IR fs laser cauterization to establish ectopic anchoring sites (refer to[2,4,63,64] and Methods). After generating two fixed sites at asymmetric positions along the AP axis of the embryo, the ventral tissue contracting in between the two ectopic fixed points and the AP

midline still flattens, preserving tissue furrowing (Fig. 5f and Supplementary movie 7). Remarkably, under asymmetric boundary conditions, the ventral tissue midline now flattens along a direction that is no longer parallel to the AP axis but instead follows a line intersecting the two ectopic anchoring sites (Fig. 5f, dashed line). This shows that the position of anchoring sites defines the boundary conditions controlling the direction of AP tissue midline flattening during furrow formation.

## Discussion

Epithelial furrowing, eventually followed by tissue internalization, is a fundamental process during embryo gastrulation and neurulation. The mechanical process responsible for furrow formation is unclear. In this study, we use as model system the *Drosophila* embryo and study furrow formation of the prospective mesoderm at the onset of gastrulation. While previous work has supported the idea that forces along the lateral side of cells play a key role and that cell wedging is a necessary step in furrow formation[24,33,54], we now challenged this view by developing a computational model based on a thin ellipsoidal elastic sheet in the 3D space, backed by 3D imaging, laser-based manipulation and multidimensional image analysis. Our model is based on two key assumptions imposed for simplicity: (i) the sheet has homogeneous linear mechanical properties, with a Poisson modulus and a ratio of Young to bending moduli which are tuned to match observed in-plane deformations, and (ii) the sheet is purely elastic, therefore, viscous properties are neglected. This last assumption, while not suited for modelling mesoderm internalization (during which mesoderm cells intercalate[17]), may be suitable to model the initial rapid process of mesoderm furrowing. By imposing a ventral pre-stress proportional to MyoII distribution measured in vivo at the apical surface of cells, we show that our computational model can predict the magnitude and the dynamics both of furrow formation and of cell apical shape changes, which happen simultaneously in the model. The concordance between our prediction and the observed dynamics confirms that the viscous response is negligible compared to the tissue elastic response, presumably because the mechanical load generated by the pre-stress is rapidly increasing and does not allow relaxation to occur (see Supplementary Information). Importantly, our model shows that surface stress in a thin, curved and purely elastic sheet in the 3D space is sufficient to drive the formation of a furrow. After forming a furrow, the elastic sheet does not invaginate. This shows that surface mechanics, while being sufficient to drive the formation of a furrow, is insufficient for subsequent tissue internalization. These computational results are in agreement with the phenotype shown by acellular embryos that are able to form a ventral furrow but fail to internalize the mesodermal cells. Therefore, cytoplasmic compartmentalization, cell lateral or basal forces and cell wedging, while dispensable for furrow formation, may be necessary for the second phase of mesoderm invagination during which the cells in the furrow are internalized.

Remarkably, the dynamics of furrow formation differ from those of MyoII and of cell shape change: while the change of MyoII intensity and cell apical area are smooth, the furrow forms abruptly (as shown by our DV curvature analysis). Such abrupt dynamics are reminiscent of the mechanical process of sheet buckling. At the onset of MyoII increase, the surface area of ventral cells is strongly reduced (from 20% at time 0 to 40% at 2 min). Our model shows that, during this first phase, most of the energy corresponding to the pre-stress results in DV contraction of ventral cells and in AP tension along the ventral midline. In this way, the DV curvature of the embryo surface at the ventral midline is reduced and eventually stalls at a smaller value (i.e., the

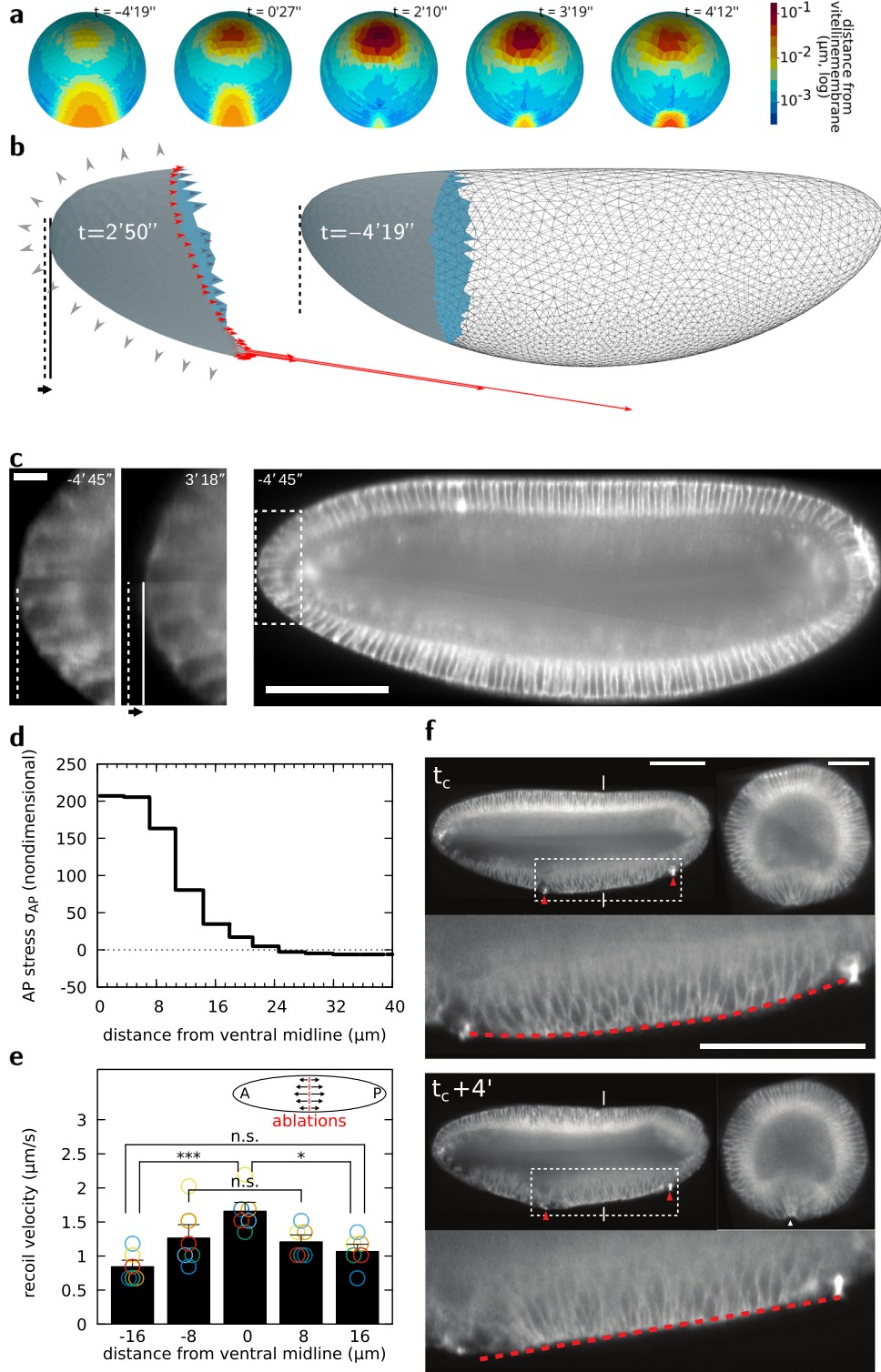

**Fig. 5 Embryo poles function as anchoring sites for ventral midline flattening and furrow formation. a** Distance map of the apical surface to the vitelline membrane at different phases of VFF. **b** Forces exerted on the poles by the rest of the tissue (red arrows), and pressure forces exerted by the incompressible cytoplasms (grey arrowheads), pole tissue deformation (compare shape of solid-filled regions) and displacement of the pole (solid and dashed lines) in simulations. **c** Digital mid-sagittal section showing inward displacement of the pole tissue during VFF. Scale bar 100 μm, zoom 5 μm. **d** Tension distribution at different DV positions from the ventral midline in simulations at $t = 0'30''$. **e** Recoil velocity distribution after DV-oriented IR fs laser ablation at different DV positions from the ventral midline. $n = 6$ embryos, data are presented as mean values ± standard deviation; the statistical test performed was Kruskal–Wallis test for multiple comparisons, $*p \leq 0.05$; $***p \leq 0.001$ and n.s. (not significant), $p > 0.05$. **f** Digital mid-sagittal and cross sections of an embryo on which two cauterizations (red arrowheads), acting as fixed points, have been performed at the ventral side (tc indicates time of cauterization). The red line indicates tissue straightening along the embryo mid-sagittal section in between the two cauterized regions. Experiment performed and result reproduced 3 times. Scale bar sagittal view 100 μm. Scale bar cross-section 50 μm.

tissue tends to flatten), whereas the AP curvature, which is initially smaller than the curvature along the DV axis, does not significantly change. This first phase is then followed by a second phase that results in an abrupt change of sign of the tissue curvature (i.e., from convex to concave) along the DV axis: this corresponds to the first appearance of the furrow that subsequently deepens over time. In our model, a new mechanical balance characterizes this second phase, during which most of the energy corresponding to the pre-stress is converted to work to bend the sheet. It is important to note that, even though the formation of the furrow is abrupt, there is no discontinuity in the sequence of tissue shapes. Since our model is free of energy dissipation, each of these shapes is at equilibrium. Thus, the control parameter (the tissue pre-stress reflecting MyoII activity) unequivocally determines the embryo surface profile. In plants, buckling has been shown to drive tissue deformation at a time scale much shorter than tissue growth[65,66]. In these cases the buckling dynamics are set by either inertial or viscous forces opposing the motion. In our model, the buckling dynamics are under the control of the active molecular mechanisms generating pre-stress during VFF.

Our model makes the prediction that the ventral furrow forms by propagating from the medial towards the polar regions of the embryo driven by a homogeneous distribution of MyoII along the ventral tissue. Our multidimensional in vivo analysis confirms furrow propagation in the absence of a MyoII wave. Furrow propagation emerges from the ventral tissue that folds sooner and faster in the central region of the mesoderm compared to regions closer to the poles, so that the furrow reaches the same absolute depth at different AP positions. This results in the flattening of the mesoderm at the ventral AP midline (i.e., at the furrow apex). Our study shows that the ventral midline, subject to the highest stress, works as a contractile string (like a 'cheese cutter wire') forming the furrow. The polar caps are pulled by the contracting ventral sheet, resisting the AP stress and therefore working as anchoring sites. The position of the polar caps imposes a ventral flattening that is parallel to the AP axis. Therefore, our study shows that embryo-scale mechanics, rather than local cell shape changes, is the key mechanism driving a buckling of the epithelial surface, forming a furrow which propagates and initiates mesoderm invagination. Models based on mesoderm cell-wedging would instead fail to reproduce furrow propagation in the absence of a MyoII wave. Furrow propagation is a process that is often reported during epithelial tube formation undergoing wrapping (e.g., during neural tube formation[67]). Propagation of a fold can be simply triggered by the propagation of a signalling or mechanical wave along a determined path. Here we show that, even in absence of triggering waves, the process of furrow propagation can emerge from the long-range mechanical interaction between epithelial tissues.

Mechanical balance results from the interaction among the material units constituting the embryo. Since mechanical interactions span the entire embryo and are much faster (almost instantaneous) compared to the establishment of morphogen gradients or mechano-chemical waves, they account for long-range tissue interaction and eventual coordination. A 2D model of a 3D system, while potentially effective to provide some level of understanding with less computational resources and programming complexity, may bias the understanding or miss key aspects of a process that emerges from the global mechanical balance and the intrinsic 3D physical nature of the system. While in the past, the choice of performing 2D imaging, analysis and computational modelling was often dictated by technology limitations, we are now at an exciting time when we can study morphogenesis by performing multidimensional image analysis and by bridging scales from the embryo to the cell and back. New imaging technology provides a synthetic view of the coordination of tissues at the scale of the whole embryo with subcellular resolution[1–4,68] while computational modelling allows the real embryo geometry to be accounted for[5,7,8,69], opening new avenues to unravel the physical principles governing morphogenesis.

## Methods

**Mutants and fly stocks.** All fly stocks were reared on standard corn meal in a 25 °C incubator. For collecting embryos for live imaging, flies were kept in cages and were allowed to deposit eggs on agar plates. ubi:Gap43::mCherry; klar flies[17] that express a plasma-membrane targeted protein fused to mCherry fluorophore were used for 3D segmentation of mesoderm cells during folding. In all other experiments to monitor membranes in live embryos were collected from ubi::Gap43::mCherry (I) fly stock. For laser ablation experiments, embryos were collected from Sqh::mCherry; Spider::GFP fly stock, in which the Myosin II regulatory light chain (MRLC) (Spaghetti Squash, Sqh, in Drosophila) is labelled by mCherry and membrane by Spider (Gilgamesh) fused to GFP. For wild-type embryo cross-sections depicting MyoII localization, embryos were collected from fly stock expressing Sqh::mCherry. For cross-sections of acellular mutant embryos showing MyoII localization, embryos were collected from Δ halo slam dunk[1]/ CyO, SqhGFP fly stock. We also analysed movies of sqh^AX3; sqh-GFP; GAP43-mCherry[70], see below "Analysis of confocal movies" for more details.

**Actomyosin meshwork ablation.** Apical actomyosin meshwork was ablated using a tunable femtosecond-pulsed infrared laser (IR fs, MaiTai) mounted on a Zeiss LSM 780 NLO confocal microscope tuned at 950 nm. Experiments were performed using a 40 × 1.2 NA objective, and the bleach mode on the Zeiss Zen software (140 mW laser power at the focal plane, single iteration, 1 μs pixel dwell, 1 s frame rate). For segmented ablations and experiments to probe the necessity of apical actomyosin contractility for furrowing, only single ablations across the ventral tissue were performed. Cross-section curvature analyses were performed along the ablated zone. For experiments to block furrow invagination, a grid pattern of iterative ablations of the apical actomyosin network were performed every time the network was recovering to disfavour actomyosin network restoration that is otherwise unstoppable.

**In toto embryo imaging and laser cauterization.** Embryos were staged and dechorionated in bleach before mounting. Embryos were mounted in a glass capillary filled with 0.5% gelrite, with their long axis parallel to the capillary. A small portion of gelrite containing the embryo was then pushed out from the capillary. The embryo was imaged on a MuVi SPIM (Luxendo, Bruker) equipped with Olympus 20 × 1.0 NA objectives and 488 and 594 nm lasers. Z-stacks were acquired with a step-size of 1 μm and during each acquisition embryos were imaged in two opposing orthogonal views (0°-dorsal-ventral view, 90°-lateral view). Thus, for every single time point, four 3D stacks were recorded. Fusion of four stacks was obtained by using Matlab[4] resulting in a final isotropic pixel resolution of 0.29 μm. Laser cauterization was performed by coupling a femtosecond 920 nm laser (Alcor2, SPARK LASERS) to MuVi SPIM and by following a similar protocol as presented in De Medeiros, G. et al.[63].

**Quantifications in MuVi SPIM data**

*Measuring furrow propagation along the AP axis.* Cross-sections were digitally made 50 μm apart along the AP axis. Furrow depth at different AP positions was measured using a dedicated point-picker ImageJ macro.

*Measuring anterior pole distance.* Mid-sagittal sections were digitally extracted from the time-lapses and the position of the vitelline membrane at the anterior pole and the apical position of the anterior-most blastoderm cell were recorded using the point-picker plugin. The relative apical position of the anterior-most blastoderm cell was calculated with respect to the position of the vitelline membrane at the anterior pole and this was plotted over time.

*Measuring tissue shortening.* Digital mid-cross and sagittal sections were obtained using ImageJ to measure tissue shortening along the DV and AP axes. Different embryos were aligned in time by keeping $t0$ as the frame at which cell apical area reduces to 20%. To measure tissue shortening along the AP axis, an identity cell was fixed at the half-length along the mid-sagittal section. Cells located 100 μm anteriorly and posteriorly from this point were marked and their distance from the identity cell was measured over time along the embryo surface. To measure tissue shortening along the DV axis, an identity cell was fixed at the midline of cross-sectional images at the ventral side. The distance along the embryo surface of the two cells located on opposite sides four cells away from the identity cell was then measured over time.

*Recoil velocity.* To measure actomyosin network recoil after ablation, the point-picker plugin was used to follow the cut end. Distance moved by the network after

ablation was plotted against time and the maximum recoil velocity was measured by calculating the first derivative.

*Curvature analysis.* An ImageJ macro was developed to measure the radius ($R$) of a circle generated from three consecutive points. The curvature was calculated as $1/R$ and was plotted against time. Curvature of the mesoderm tissue was measured both along the AP and DV axes. A convex curvature is given by a positive and a concave curvature is given by a negative value.

*3D image segmentation and analysis.* Mesodermal cells were segmented and tracked by inter-registration based on iterative projections of segmentations from one time point to another using the ASTEC algorithm[71]. Morphological data were extracted and analysed using Python. ImageJ dedicated macros were used for image treatment and the 3D Viewer plugin for rendering[72].

## Analysis of confocal movies

*Cell tracking.* Apical shapes and *sqh*-GFP intensities were calculated for mesoderm cells during gastrulation for three wild-type ($sqh^{AX3}sqh-GFP$; $GAP43-mCherry$) embryos imaged and analysed in[70]. Embryo movies SG_1, SG_3 and SG_4, which started prior to apical constriction and with similar overall *sqh*-GFP intensities, were used for this study. Confocal live imaging, with confocal stacks captured every 30 s, and cell tracking are as described in[70], with the evaluation of cell apical dimensions taking into account the local inclination of the embryo[73]. Mesoderm cells were identified as all cells that did not remain on the surface of the embryo after gastrulation. For every tracked mesoderm cell in each movie frame, apical cell area, AP and DV lengths of the best-fit ellipse to the cell shape, distance of the cell centroid from the ventral mid-line and average apical *sqh*-GFP intensity were extracted. Cells located at a given DV distance to ventral midline at $t = -4$ min were then pooled together within AP-oriented rows and we studied the time evolution of the variables of interest for each row. Some cells cannot be tracked over the whole duration of the embryo movie, either because they are initially partly outside the field or because in variations in the fluorescence of the *GAP43* membrane signal. Variations of the variable of interest of each tracked cell between two consecutive frames were calculated for each cell tracked in those frames, then averaged over cell row. The normalized variable for the row is defined as being 1 in the first frame and then varying according to this average variation, which reflects the variations in size and fluorescence intensity of the row in the course of the whole movie.

*MyoII signal analysis and pre-stress.* The MyoII signal can usefully be decomposed into a constant signal $I_c$ (equal to 4 arb) and a signal $I_i$ that depends on the row $i$ considered and time, that we named myosin excess fluorescence. In Fig. 1g we plotted $I_i(t)/I_0(t)$ as a function of $i$ for all the time points (colour-coded). These myosin profiles are well conserved across the VFF process, so that we further decomposed the MyoII signal as the product of a row dependent profile function (plotted in Fig. 1g) and a time dependent function (fitted in Eq. (1), see Fig. 3a). We used this fitted excess fluorescence to prescribe pre-stress $\sigma_a$ in simulations.

## Statistics and reproducibility
Number of reproduction of each experiment is specified in the caption of the corresponding figure. No statistical method was used to predetermine sample size. No data were excluded from the analyses, although only the confocal microscopy embryo movies analysed in ref. [70] which extended sufficiently early in time and had appropriate fluorescence signal level were included, as stated above. The experiments were not randomized and the Investigators were not blinded to allocation during experiments and outcome assessment.

## Time synchronization
Time 0 was set as the instant when cells within the 5 rows closest to the midline had their area reduced by 20% compared to pre-gastrulation, allowing to synchronize all embryos, whether imaged with SPIM or confocal. Interpolation between datasets and time derivatives were calculated by LOESS moving window technique with dedicated software[74]. The dynamics of development was found to be different between MuVi SPIM and confocal experiments due to differences in temperature. It was found that area reduction in confocal experiments was taking place at a twice slower rate than in MuVi SPIM, this factor was applied in Fig. 3a and for all subsequent confocal experiments. This corresponds to $T = 20$ min in Eq. (1).

## Numerical simulations
The geometry of the elastic surface used to construct the initial finite element mesh, as well as the volume constraint and constraint imposed by vitelline are decribed in Supplementary Information. A linear elastic model is then solved with the Surface Evolver software[52] with increasing pre-strain for the ventral region highlighted in Fig. 1f, as described in Supplementary Information.

**Reporting summary**. Further information on research design is available in the Nature Research Reporting Summary linked to this article.

## Data availability
The data that support the findings of this study are available in Zenodo with the identifier https://doi.org/10.5281/zenodo.6043523.

## Code availability
The scripts that were used to process the data and perform the numerical simulations in this study are available in Zenodo with the identifier https://doi.org/10.5281/zenodo.6043523.

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

## Acknowledgements

A.J., J.E. and M.R. acknowledge that this research was supported in part by the National Science Foundation under Grant No. NSF PHY-1748958 while in KITP at UC Santa Barbara. M.R. thanks the company Luxendo Bruker and SPARK LASERS for fruitful collaborations. A.J., B.D, A.P. and M.R. thank PRISM imaging facility for technical support. This work was supported by the French government through the UCAJEDI Investments for the Future project managed by the National Research Agency (ANR-15-IDEX-01), the Investments for the Future LABEX SIGNALIFE (ANR-11-LABX-0028-01), the ATIP-Avenir programme of the CNRS, the Human Frontier Science Programme (CDA00027/2017-C) and the Region SUD PACA Research and ERC Booster programmes. G.B., C.L. and B.S. acknowledge financial support of a Wellcome Trust Investigator Awards 099234/Z/12/Z and 207553/Z/17/Z. J.F., A.T., J.E., C.Q. and P.M. acknowledge financial support from the European Community's Seventh Framework Programme (FP7/2007-2013) ERC Grant Agreement Bubbleboost

no. 614655, and are members of GDR 3570 MecaBio and GDR 2108 AQV of CNRS. J.E. was additionally supported by IRS "AnisoTiss" of Idex Univ. Grenoble Alpes. G.B., C.L., B.S. and J.E. benefited from a PICS CNRS travel grant and ANR-11-LABX-0030 'Tec21' grant. The computations were performed using the Cactus cluster of the CIMENT infrastructure, supported by the Rhône-Alpes region (GRANT CPER07-13 CIRA). The authors thank Philippe Beys who manages the cluster, and K. Brakke for developing and maintaining the Surface Evolver software as well as invaluable interactions during this work.

## Author contributions

J.E., P.M., C.Q. and M.R. conceived the project. M.R. planned the in vivo experiments. B.S. planned complementary confocal experiments. J.E., P.M. and C.Q. planned the modelling and computational approach. J.F., A.T. and C.Q. performed the computations. J.F., C.Q., P.M. and J.E. analysed the computational results. A.J., B.D., A.P. and M.R. performed the laser-based manipulations, light sheet experiments, data processing and analysis. C.M.L., G.B.B. and J.E. performed the analysis on complementary confocal images. G.M. provided support on the 3D data processing. J.E. and M.R. wrote the manuscript. All authors commented on the manuscript.

## Competing interests

The authors declare no competing interests.
