## [Peer Review File · Nature Communications]

Embryo-scale epithelial buckling forms a propagating furrow that initiates gastrulationREVIEWER COMMENTS

Reviewer #1 (Remarks to the Author):

The authors investigate the mechanics of furrow formation during the invagination of the mesoderm in *Drosophila* embryos. The authors develop a continuous elastic model of furrow formation. The model predicts that the furrow forms medially and then spreads anterior and posteriorly, which is indeed the case in vivo. Using multi-view SPIM, the authors find that the ventral surface of the mesoderm flattens along the AP axis as the tissue folds, an observation predicted by the model. Furthermore, the model suggests that the polar tissues may serve as anchors that resist AP stress and flatten the mesoderm, thus promoting furrowing. The authors validate this prediction by generating ectopic anchor points based on laser-induced cauterization, leading to flattening that connects the two anchor points.

This is a well conducted study, with beautiful imaging and solid mathematical modelling. The first three figures are not too surprising in the context of the existing literature (which in a few key cases is not referenced), and the results in Figure 5 are interesting, but somewhat qualitative. I propose to address the following points:

MAJOR

1. In their Discussion (page 15) the authors write: "Importantly, our model shows that stress at the apical surface of a thin curved elastic sheet in the 3D space is sufficient to drive the formation of a furrow." But a mathematical model does not show anything: it provides conditions consistent with the in vivo observations that then need to be tested experimentally. In addition to this, didn't the Martin and Dunkel labs generate a similar prediction based on a similar 3D continuum elastic model (Heer et al., *Development* 2017)? The authors should tone down their claim and discuss how their findings are different from those in the previous model.
2. Related to the previous point, the authors fail to cite recent experimental work by the DeRenzis and Glotzer labs that directly demonstrate that apical myosin activation (by optogenetic activation of Rho1) is sufficient for fold formation. At the very least the previous work should be referenced (see Izquierdo, Quinkler and DeRenzis, *Nature Communications*, 2020; and Rich, Fehon and Glotzer, *eLife* 2020), and the authors should address how their findings differ from previous ones or what is the new contribution of their work.
3. In their modelling of the reasons why cells in rows 6-9 stretch, the authors fail to consider/cite/discuss recent work from the Martin lab showing that those cells display a decrease in cortical F-actin, which reduces their resistance to deformation. In light of existing experimental and molecular evidence that explains whether cells constrict, stretch or maintain their shape (Denk-Lobnig et al., *Development* 2021), I am not sure how much the model adds here. This should be clearly explained, as well as discussed in the context of the previous results.
4. Figure 5: the results of the cauterization experiment are qualitative. Can the authors demonstrate that asymmetric cauterization changes the distribution of tension shown in 5e, shifting the peak towards the new anchor point? Also, is there a consequence of the change in anchors for the invagination of the mesoderm?
5. Based on the "cheese cutter wire" model proposed by the authors in the Discussion, one would hypothesize that cauterization on both sides of the midline would lead to the formation of a DV oriented furrow as mesoderm cells try to constrict. Is that the case?

MINOR

1. Figure 3a: the error band around the MuVi SPIM data is not visible (there should be one).
2. Figure 3a, b: the agreement between model and in vivo data is nice, but I wonder if it could be

improved if the localized pre-stress used in the model were set using the in vivo myosin values (Figure 3a dotted red line) rather than a mathematical fit of those values (Eq. 1)?

3. Figure 5c: I cannot see the vitelline membrane. The authors should show a transmitted light image or use the autofluorescence from the vitelline membrane to show the separation of polar tissues from the vitelline envelope.

4. Figure 5e: I thought the authors were discussing the force distribution along the AP axis with respect to distance from the midline. But according to the legend, these are cuts at different DV positions? Is that correct or should it be different AP positions?

5. Figure 4: The idea that cell shortening is necessary for folding of the furrow has recently been investigated by the De Renzis lab (Krueger et al., EMBOJ 2018). Are flattening and cell shortening related, and if so, are their molecular mechanisms the same?

TYPOS

1. Page 9: "Consistently with our model" should be "Consistent with our model".

Reviewer #2 (Remarks to the Author):

Fierling et al Nature Communications

This manuscript, by Fierling et al addresses the forces for morphogenesis during the formation of the ventral furrow during *Drosophila* gastrulation. The authors use a combination 3D computational modeling and in toto embryo image analysis and manipulation to report that embryo-scale force balance of the tissue surface rather than cell-autonomous shape changes are necessary and sufficient to drive buckling of the embryo surface to form a furrow that propagates and initiates embryo gastrulation. Of special interest is that 3D modeling cannot be mimicked by a 2D treatment. They note that their model generates the furrow, but not ingression of the mesodermal cells. Ventral furrow formation is a well-studied model system and should be of interested to a significant fraction of the readership of Nature Communication.

Unfortunately, this is a poorly organized manuscript that is hard to follow and very incompletely describes experimental manipulations. In particular the paragraphs are long and rambling, with topic sentences that fail to cover paragraph content. The authors need to revise with shorter paragraphs that focus on a single topic and topic sentences that adequately describe what the paragraph is about. They should be more explicit about informing the reader of the take home message of each of their observations as concluding sentences in the paragraphs that make them or as in a summary paragraph in each section. Moreover, manuscripts such as this one have a unique opportunity to be pedagogical: for maximum impact, the modelers need to educate readers who are more biological on precisely how their model maps onto the biology, where there are simplifying assumptions and how the model and its math can be interpreted to understand the spatially and temporally varying mechanical properties that ultimately specify morphogenesis.

Specifics:

The authors report the use of femtosecond IR laser to sever the actomyosin networks that drive apical contraction. They report that following dissection of the network, the network recoils and the cell apical surface dilates, citing Figure 1. They state that the network finally recovers, restoring apical contraction forces and cell apical constriction. The authors do not do an adequate job of presenting

their data. To me it looks as if the laser cut is a line that is perpendicular to the long axis of the embryo. Unfortunately, the authors never tell us (in the text, in the figure legend or in the methods), the orientation of the cut and how long it was. In addition, the authors assert that the two photon severing of the actomyosin network leaves the membrane intact. But the panels shown as Fig 1a "membrane" has a lesion in membrane fluorescence channel that looks as severe (and almost identical to) the lesion in the myosin channel. I suppose it is possible that there is bleaching of membrane fluorescence without damaging the membrane, but the subsequent pattern of recoil seems to suggest against that interpretation. The authors need to explicitly tell the reader where the cuts are made and why the membrane channel looks so similar to the myosin channel if the effect of the laser are to sever only the actomyosin network. Other features of the cuts suggest that the authors' interpretation may be correct, but the behavior of the membrane warrants further explanation. Further, while this study is focused on apical constriction, without a junctional marker like fluorescent cadherin or beta catenin, it's virtually impossible to discern what the cell shapes are during recoil and recovery. Fig 1b makes me think that indeed the cuts in Fig 1a were parallel to the anterior posterior axis of the embryo, at the dorsal midline. Again, why do the images in 1a look as if they are cutting perpendicular to the long axis of the embryo? Why were just 4 embryos analyzed in Fig 1C (presumably 4 embryos were analyzed, each embryo at the three different time points). I don't understand the time stamps on the panels in Fig 1a and the authors don't tell us what they mean in the legend. In certain cases they "synchronize" time when cells contract by a certain percentage, but it is not clear they have done so here. At least use the legend to refer to the methods if that is where time stamps are described. Why don't they simply use time 0 as the time of the cut and let us know what the embryo looked like during recoil and complete recovery (the authors did not take the embryo followed in Fig 1a to full recovery). The authors don't tell us in the legend what the colored patches in Fig 1e or f are supposed to represent. Contrary to the authors assertions, the actomyosin networks are not recovered after 10 sec. The authors do not describe how they calculated "Normalized myosin excess fluorescence" which is essential for the reader to understand the magnitude of the myosin intensity changes being measured.

The authors next sequentially ablate the actomyosin networks to ask what happens if they don't allow the networks to recover. They find that furrowing and gastrulation is inhibited. It is not clear why they use such a large array to prevent furrowing and again, why changes in fluorescence from the membrane channel is so similar to the myosin channel. The authors are probably correct in concluding that forces (presumably cell autonomous forces) generated by actomyosin networks are necessary for both furrow formation and subsequent tissue invagination. However, for the reasons described above, the data is not presented in a way that is sufficient to support that conclusion. Moreover, for reasons cited by the authors, this finding is not terribly surprising.

The authors next introduce slam dunk embryos that fail to cellularize but still make furrows. With no lateral and basal membranes/cellular structures, they argue this indicates that everything necessary for furrow formation must be driven by apical forces and build an elastic model for forces at the cell surface. They avoid the inclusion of viscosity in the model: they make the argument based on the observation that the relaxation time for the embryonic epithelium is 1 minute and furrow formation is considerably longer, and that the load generated by myosin is constantly increases. Especially for readers who are primarily biologists, the rationale for this conclusion should be more fully developed, either with a few sentences, with references here, or more extensively in the supplement.

The authors describe their continuum model for the surface of the embryo, and site previous work, confirmed here on the distribution of apical myosin in the mesoderm. They next describe pre-strain that is proportional to myosin intensity. Unfortunately, explicit description of pre-strain is buried and deserves a paragraph of its own. It is not clear why pre-strain should be proportional to myosin intensity (or rather "normalized myosin excess fluorescence") and that assertion should be developed more fully and explicitly. These details are important for full appreciation of the model.

The authors next address their overall model, how tension anisotropy emerges from tissue and

embryo geometry.

Page 8, Line 11. It is not clear what "decomposing the length of the contracting ventral tissue" is. The authors need to explain this more explicitly to keep the reader on board. In this paragraph, the authors conclude that "the shape anisotropy of the system would thus explain why the surrounding tissue appears more difficult to deform along the AP than the DV direction even though the mechanical properties of the entire tissue surrounding the mesoderm are imposed to be the same." This is a very interesting observation, and the next two paragraphs are designed to evaluate the consequences of these observations. They should be able to do this more explicitly.

Page 8, Line 20. Do the authors really mean to say that "...actomyosin networks anchored to stiff boundaries which oppose resistance to deformation...". Don't they mean "...oppose deformation..."?

Page 8, Line 27. The authors introduce "the trace of stress tensors" and "principle stresses, which are the eigenvalues of the stress tensor." Again, the authors should present their findings in a more pedagogical fashion so that biologists can readily interpret their findings. This would require more extensive treatments of "traces of stress tensors", principle stresses and eigenvalues of the stress tensor, probably in the supplement. For their model and findings to be most useful, they will be used by biologists to design new experimental tests that verify, modify or refute the model.

Fig. 3a makes a compelling case for the model providing a good fit for the biological data.

Page 10, Lines 8-10. The authors state "Remarkably, the 2D elastic sheet forms a buckle resulting in a furrow along the long axis of the 3D ellipsoid in the region under pre-stress (Fig. 4a and Supplementary Fig. 3a). This shows that forces applied at the surface of an ellipsoidal 3D shape can be sufficient to drive the formation of a furrow." The authors should expand on this to state exactly where the forces that drive buckling are located, and what their nature is. Subsequent analysis of in toto tissue movements is compelling, but the authors leave the reader to speculate as to the nature of the forces that are driving buckling (or they believe they are so obvious as to not require explicit description). Much later (page 12, line 12) the authors tell us that there is a contractile string that drives furrow formation. Are other forces involved? Is there a connection to cell autonomous forces in the cells that are in the furrow. The authors add biological test of this by cauterizing new anchors for the AP axis and show results consistent with their model.

Ultimately, the authors do not address the relationship between furrow formation and invagination of the mesoderm and the convergent extension that begins to drive germ band extension. Are furrow formation, tissue internalization completely separable events in time and space? If so, the authors need to state as much, perhaps in more than one place (e.g., in the Introduction and the Discussion). If not they need to tell us how their model for furrowing might be impacted by those other processes.

Some additional, random (or reiterated notes).

The description for Fig 1 is inadequate

In Figure legend 2, the authors describe panel e before panel d.

Reviewer #3 (Remarks to the Author):

This manuscript is globally correct but rather incremental. It convincingly shows that buckling is sufficient to explain the observations. However, buckling is not by itself a new mechanism to explain folding. In addition, as is well known, and as is shown in the authors' previous Fig 1 of Ref 19 (Rauzi Biophys J 2013), several models can explain the same phenotype. Here, the current manuscript (despite its claim, e.g. in the abstract), does not convincingly show that buckling is really necessary to explain the phenotype. It rather shows that apical contraction is necessary to drive furrow formation.

I am not sure that, after revision, it will be of interest for Nat Comm. I recommend to submit it to a more specialised journal.

Revisions :

Use consistent vocabulary. e.g. : laser dissection / ablation / manipulation (p2L20, Fig 1a, Fig 1c, p5L6, etc).

Each fig should be called in order (e.g.: P7L18, p7L21, p7L25), with a sentence which corresponds to the actual content of the figure (e.g.: p8L10, p9L27), and have an adequate title (eg should Fig 2 title be "Surface Evolver simulations" ? SuppFigs lack titles) ; make clear to the reader what is experimental and what is the result of simulations.

Clarify (p12L8-9, p13L2, p15L24-27, p15L30-31, p16L5-6), remove redundancies (e.g.: p5L7=p5L10), correct typos ("Utricularia trap" missing p23L27).

The box should be cut in two boxes. A first box on stress and strain (currently part "b"). A second box on furrowing etc (currently part "a"). Both should be more legible.

Compare better with authors' previous papers, notably with Ref 19 (Rauzi Biophys J 2013) and Ref 5 (Dicko PLoSCB 2017).

Also, note the preprint : bioRxiv preprint doi: <https://doi.org/10.1101/2021.09.16.460711>

REVIEWER COMMENTS

Reviewer #1 (Remarks to the Author):

The authors investigate the mechanics of furrow formation during the invagination of the mesoderm in Drosophila embryos. The authors develop a continuous elastic model of furrow formation. The model predicts that the furrow forms medially and then spreads anterior and posteriorly, which is indeed the case in vivo. Using multi-view SPIM, the authors find that the ventral surface of the mesoderm flattens along the AP axis as the tissue folds, an observation predicted by the model. Furthermore, the model suggests that the polar tissues may serve as anchors that resist AP stress and flatten the mesoderm, thus promoting furrowing. The authors validate this prediction by generating ectopic anchor points based on laser-induced cauterization, leading to flattening that connects the two anchor points.

This is a well conducted study, with beautiful imaging and solid mathematical modelling. The first three figures are not too surprising in the context of the existing literature (which in a few key cases is not referenced), and the results in Figure 5 are interesting, but somewhat qualitative.

I propose to address the following points:

MAJOR

1. In their Discussion (page 15) the authors write: "Importantly, our model shows that stress at the apical surface of a thin curved elastic sheet in the 3D space is sufficient to drive the formation of a furrow." But a mathematical model does not show anything: it

provides conditions consistent with the in vivo observations that then need to be tested experimentally.

In the sentence pointed out by Reviewer#1 we do not aim at making a statement regarding the embryo biophysics, but rather on the purely physical behaviour of an elastic sheet. This sentence and the following two summarise some of our numerical findings, which we then claim to be “*in agreement with the phenotype shown by acellular embryos*”. The issue may originate from the usage of the anatomical word “apical surface” in that context. We have changed the term “apical” in sentences which refer only to the model.

To make this point clearer we now write on p15: “*Importantly, our model shows that surface stress on a thin, curved and purely elastic sheet in the 3D space is sufficient to drive the formation of a furrow.*”

In addition to this, didn't the Martin and Dunkel labs generate a similar prediction based on a similar 3D continuum elastic model (Heer et al., Development 2017)? The authors should tone down their claim and discuss how their findings are different from those in the previous model.

We thank Reviewer#1 for pointing out the modelling study of Heer et al 2017, which we now cite also in the introduction and discussion. The Heer et al. model is different from our model in both (1) the modelling parameters and in (2) the resulting output. Here follows the rationale:

- 1 The Heer model is different from our model and is more similar to previous models in terms of modelling parameters. In the Heer model there are forces along the apico-basal direction. In fact, Heer et al. impose constant thickness: “*we assume that the cell is significantly stiffer against vertical compression than against horizontal, and treat the cell height as a fixed quantity h* ” (above eqn 2 in their SI Text). Cell stiffness is associated to a force-deformation relation in which the deformation is imposed to be equal to 0. This results in a reaction force exactly balancing any force tending to modify h . In other terms, Heer et al consider their elastic shell model to be based on the physics of two superimposed layers having different properties. This results in apical-basal differential tension driving torque which is the same physic principle upon which previous wedging models (reviewed in Rauzi et al 2013) are based. Our model is instead based on the mechanics of one single thin sheet (meaning that there is no “inner” and “outer” sheet) and is consequently torque-free (i.e., no resulting forces along the cell apico-basal direction).
- 2 The Heer model is different from our model and more similar to previous models in terms of resulting output. Our model can predict key *in vivo* embryo shape changes (e.g., AP tissue flattening along the ventral midline and furrow propagation) and can predict phenotypes observed in mutated embryos lacking lateral and basal cell sides (slam-dunk- embryos). The Heer et al. and all previous 3D models fail to predict this key features of VFF, which had not been documented experimentally before.

2. Related to the previous point, the authors fail to cite recent experimental work by the DeRenzis and Glotzer labs that directly demonstrate that apical myosin activation (by optogenetic activation of Rho1) is sufficient for fold formation. At the very least the previous work should be referenced (see Izquierdo, Quinkler and DeRenzis, Nature

Communications, 2020; and Rich, Fehon and Glotzer, eLife 2020), and the authors should address how their findings differ from previous ones or what is the new contribution of their work.

We thank Reviewer#1 for this insightful comment. We are aware of the work from the De Renzis lab and the Glotzer lab, which we now cite and discuss. We believe that the furrowing they observe corresponds to a mechanism that differs from the initial furrow formation that is the focus of our paper. Indeed, the basal and lateral sides could still passively contribute in the deformations that the De Renzis and Glotzer lab report (in line with the computational model by O. Polyakov et al. 2014). This would also explain why photo-induced furrows do not look like wild type VFF: while the former result in a bend along both the DV and the AP axes, the latter results in a bend along DV but a flattening along AP.

We are now discussing this in the second section of Results: *“Optogenetic MyoII activation in the vicinity of the apical surface has shown that furrowing can be achieved by triggering active contractility [Izquierdo et al, 2018; Rich et al 2020]. However, passive basolateral forces could also be at play, resulting from the elastic modulus associated to cell lengthening [Polyakov et al, 2014] or to a constant cell length being imposed [Heer et al, 2017]”*

3. In their modelling of the reasons why cells in rows 6-9 stretch, the authors fail to consider/cite/discuss recent work from the Martin lab showing that those cells display a decrease in cortical F-actin, which reduces their resistance to deformation. In light of existing experimental and molecular evidence that explains whether cells constrict, stretch or maintain their shape (Denk-Lobnig et al., Development 2021), I am not sure how much the model adds here. This should be clearly explained, as well as discussed in the context of the previous results.

We thank Reviewer#1 for pointing out the recent work of the Martin lab. We agree with Reviewer#1 that the F-actin pattern described by Denk-Lobnig can further contribute to the global mechanical balance resulting in the observed cell stretch. We have now added a reference to their work: *“This effect could be additionally enhanced by local modulations of the mechanical properties of filamentous actin within the mesoderm (Denk Lobnig et al 2021), resulting in an even larger stretch in vivo than predicted by the model.”*

4. Figure 5: the results of the cauterization experiment are qualitative. Can the authors demonstrate that asymmetric cauterization changes the distribution of tension shown in 5e, shifting the peak towards the new anchor point?

We believe there is a bit of confusion on this point. We are not expecting to have any quantitative change of the AP tension gradient along the DV axis as shown in 5e. The only expected effect is that the flattening happens along the ectopic anchoring sites and that furrowing still takes place along the DV axis. This is shown in Fig. 5f and in Supplementary Movie 7. AP tension gradient along the DV axis is not affected since we are not manipulating the MyoII DV gradient. We have now added an explanatory insert in Fig 5e to make it clearer and also performed additional quantifications in Fig S3c to instead corroborate the idea that AP tension is the same at different AP positions.

Also, is there a consequence of the change in anchors for the invagination of the mesoderm?

Yes, the change in anchor from the ‘floating polar caps’ to the ‘fixed cauterized regions’ can eventually induce a change in the dynamics of mesoderm internalization (the process taking place after ventral furrow formation). With fixed anchoring sites the mesoderm stalls for some time and in some cases it is still capable to internalize, forming a bend not only along DV but also along the AP direction. We believe that tissue internalization is a more complex process that would eventually relay on multiple mechanisms as for instance embryo-scale buckling in combination with cellular torque (as mentioned in the discussion). Nevertheless, the process of tissue invagination (after ventral fold formation) goes beyond the scope of our work here and it deserves a dedicated experimental study.

5. Based on the "cheese cutter wire" model proposed by the authors in the Discussion, one would hypothesize that cauterization on both sides of the midline would lead to the formation of a DV oriented furrow as mesoderm cells try to constrict. Is that the case?

In order to obtain a “cheese cutter wire” is necessary to first form “a wire”, that is, a narrow stripe of constricting cells. The wire, in the WT and our cauterization experiment, is represented by the stripe of constricting cells that is long along AP and narrow along DV. Cauterizing along the DV axis would therefore not form a “cheese cutter wire” because there is no mechanism for a convergence motion towards the line between the cauterized loci. DV cauterization was performed in Rauzi et al. 2015. This impedes lateral tissue movement blocking ventral folding.

MINOR

1. Figure 3a: the error band around the MuVi SPIM data is not visible (there should be one).

We thank Reviewer#1 for pointing this out. We now corrected this.

2. Figure 3a, b: the agreement between model and in vivo data is nice, but I wonder if it could be improved if the localized pre-stress used in the model were set using the in vivo myosin values (Figure 3a dotted red line) rather than a mathematical fit of those values (Eq. 1)?

It may improve, although the large inter-embryo variations would probably require to adjust and compare separately each specific case. That said, with this work our goal is to implement the simplest and general model that can recapitulate the main features of the system to eventually highlight novel driving mechanisms, rather than exactly mimicking fine cell shape dynamics.

3. Figure 5c: I cannot see the vitelline membrane. The authors should show a transmitted light image or use the autofluorescence from the vitelline membrane to show the separation of polar tissues from the vitelline envelope.

We now show this by exploiting vitelline membrane auto-fluorescence at 488 nm. This is now presented in panel Fig.S3g with further quantifications.

4. Figure 5e: I thought the authors were discussing the force distribution along the AP axis with respect to distance from the midline. But according to the legend, these are cuts at different DV positions? Is that correct or should it be different AP positions?

We agree with Reviewer#1 that this point has generated confusion. The tension in Fig.5 d and e is along AP at different DV positions. The point is indeed to show that the midline has the highest AP tension to function as a “cheese cutter wire” (as represented in Fig.5b, red arrows). We have now added an explanatory insert in the plot to make it clearer and avoid confusion.

5. Figure 4: The idea that cell shortening is necessary for folding of the furrow has recently been investigated by the De Renzis lab (Krueger et al., EMBOJ 2018). Are flattening and cell shortening related, and if so, are their molecular mechanisms the same?

We thank Reviewer#1 for this interesting question. We think that AP tissue flattening is not related to ventral cell shortening. The latter we think to be more related to apical-basal tension difference (Krueger et al.) driving bending moment (i.e., cellular torque). The flattening of the ventral tissue results instead in a global movement of the prospective mesoderm cells towards the interior of the embryo. AP ventral flattening is concomitant with initial apical constriction that eventually correlates with ventral cell lengthening (Gelbart et al. 2012).

TYPOS

1. Page 9: "Consistently with our model" should be "Consistent with our model".

We thank Reviewer#1 for pointing this out. This is now corrected.

Reviewer #2 (Remarks to the Author):

This manuscript, by Fierling et al addresses the forces for morphogenesis during the formation of the ventral furrow during Drosophila gastrulation. The authors use a combination 3D computational modeling and in toto embryo image analysis and manipulation to report that embryo-scale force balance of the tissue surface rather than cell-autonomous shape changes are necessary and sufficient to drive buckling of the embryo surface to form a furrow that propagates and initiates embryo gastrulation. Of special interest is that 3D modeling cannot be mimicked by a 2D treatment. They note that their model generates the furrow, but not ingression of the mesodermal cells. Ventral furrow formation is a well-studied model system and should be of interested to a significant fraction of the readership of Nature Communication.

Unfortunately, this is a poorly organized manuscript that is hard to follow and very incompletely describes experimental manipulations. In particular the paragraphs are long and rambling, with topic sentences that fail to cover paragraph content. The authors need to revise with shorter paragraphs that focus on a single topic and topic sentences that adequately describe what the paragraph is about. They should be more explicit about informing the reader of the take home message of each of their observations as concluding sentences in the paragraphs that make them or as in a summary paragraph in each section.

Moreover, manuscripts such as this one have a unique opportunity to be pedagogical: for maximum impact, the modelers need to educate readers who are more biological on precisely how their model maps onto the biology, where there are simplifying assumptions and how the model and its math can be interpreted to understand the spatially and temporally varying mechanical properties that ultimately specify morphogenesis.

We thank the reviewer for their comments. We have now worked on the text of the manuscript to improve its organization and readability.

Specifics:

The authors report the use of femtosecond IR laser to sever the actomyosin networks that drive apical contraction. They report that following dissection of the network, the network recoils and the cell apical surface dilates, citing Figure 1. They state that the network finally recovers, restoring apical contraction forces and cell apical constriction. The authors do not do an adequate job of presenting their data. To me it looks as if the laser cut is a line that is perpendicular to the long axis of the embryo. Unfortunately, the authors never tell us (in the text, in the figure legend or in the methods), the orientation of the cut and how long it was.

We would like to point out that Fig.1a is only intended to show that after dissection i) the actomyosin network is disrupted, ii) the membrane is preserved and that iii) the actomyosin network eventually recovers. The process of dissection and recovery happens by performing any type of ablation pattern (e.g., AP ablations, DV ablations, circular ablations, etc.). Therefore, for Fig.1a the ablation pattern *per se* is not crucial.

Nevertheless, we agree with Reviewer#2 that this should be clarified. Therefore, we now specify in Fig.1a legend that the ablation was performed along the DV axis across the ventral tissue (with anterior on the left and posterior on the right) during apical constriction.

In addition, the authors assert that the two photon severing of the actomyosin network leaves the membrane intact. But the panels shown as Fig 1a “membrane” has a lesion in membrane fluorescence channel that looks as severe (and almost identical to) the lesion in the myosin channel. I suppose it is possible that there is bleaching of membrane fluorescence without damaging the membrane, but the subsequent pattern of recoil seems to suggest against that interpretation. The authors need to explicitly tell the reader where the cuts are made and why the membrane channel looks so similar to the myosin channel if the effect of the laser are to sever only the actomyosin network. Other features of the cuts suggest that the authors’ interpretation may be correct, but the behavior of the membrane warrants further explanation.

We apologize with Reviewer#2 for not being sufficiently explicit. As mentioned in the previous point, laser cuts are made along the DV axis on the ventral side of the embryo (being anterior on the left, and posterior on the right of the image) during apical constriction. The first two panels of Fig.1a show that, while MyoII is dismantled (see absence of signal in the middle panel showing the cut), membrane is still visible and cells eventually start to dilate already just after ablation at $t=0s$ (bottom panel). During dilation, cells stretch along the AP axis (from left to right). Since the cells stretch, the density (intensity/length) of membrane fluorophore is reduced. Bleaching, as suggested by Reviewer#2, also may partially contribute additively to membrane signal local decrease. Nevertheless, it is cell dilation the major cause of membrane fluorophore density decrease (see bottom panels of Fig.1a). We now specify this in the figure legend and in the main text, “After laser dissection, the network is cut and recoils while the cell apical surface membrane is preserved and dilates.”, page 5.

Further, while this study is focused on apical constriction, without a junctional marker like fluorescent cadherin or beta catenin, it’s virtually impossible to discern what the cell shapes are during recoil and recovery. Fig 1b makes me think that indeed the cuts in Fig 1a were parallel to the anterior posterior axis of the embryo, at the dorsal midline. Again, why do the images in 1a look as if they are cutting perpendicular to the long axis of the embryo? Why were just 4 embryos analyzed in Fig 1C (presumably 4 embryos were analyzed, each embryo at the three different time points).

Membrane-attached fluorescence is an ideal marker with which to discern apical cell shapes, which remain visible throughout laser ablation movies. Membrane signal has two advantages over adherens junction-specific fluorescent tags: it tends to be more uniform than punctate E-Cadherin signal, for example, and it covers the whole cell surface permitting segmentation and analysis of 3D cell shapes.

In Fig.1b laser dissection was also performed along the DV axis across the ventral tissue as in Fig.1a. Cross-sections in Fig.1b are shown along the ablated zone where cells dilate. We now specify this in the figure legend and in the method section. As Reviewer#2 mentions, yes the analyses were done on 4 different embryos during the 4 different phases: Phase0 (t_0) before apical constriction; Phase1 (t_1) during apical constriction and MyoII recruitment; Phase2 (t_2) after apical laser dissection and Phase3 (t_3) during actomyosin recovery. The same experimental trial repeated 4 times on four embryos for the four phases provided very consistent and reproducible results.

I don’t understand the time stamps on the panels in Fig 1a and the authors don’t tell us what they mean in the legend. In certain cases they “synchronize” time when cells contract by a certain percentage, but it is not clear they have done so here. At least use the legend to refer to the methods if that is where time stamps are described. Why don’t

they simply use time 0 as the time of the cut and let us know what the embryo looked like during recoil and complete recovery (the authors did not take the embryo followed in Fig 1a to full recovery).

We agree that time in Fig.1a was very confusing. We thank Reviewer#2 to point this out. We now re-edited this panel by following Reviewer#2 suggestions.

The authors don't tell us in the legend what the colored patches in Fig 1e or f are supposed to represent.

The ventral color patch in Fig.1e indicates the ventral region of the embryo. Color in Fig.1f indicate pre-stress as shown in the panel. We now better edited the corresponding figure legend.

Contrary to the authors assertions, the actomyosin networks are not recovered after 10 sec.

We thank Reviewer#2 for pointing out this mistake. We have amended the text.

The authors do not describe how they calculated "Normalized myosin excess fluorescence" which is essential for the reader to understand the magnitude of the myosin intensity changes being measured.

We thank the Reviewer#2 for pointing out this omission. We have now added a Methods title "MyoII signal analysis and pre-stress" and added more details above and below that title.

The authors next sequentially ablate the actomyosin networks to ask what happens if they don't allow the networks to recover. They find that furrowing and gastrulation is inhibited. It is not clear why they use such a large array to prevent furrowing and again, why changes in fluorescence from the membrane channel is so similar to the myosin channel. The authors are probably correct in concluding that forces (presumably cell autonomous forces) generated by actomyosin networks are necessary for both furrow formation and subsequent tissue invagination. However, for the reasons described above, the data is not presented in a way that is sufficient to support that conclusion. Moreover, for reasons cited by the authors, this finding is not terribly surprising.

Here we want to test if apical constriction is necessary for consequent tissue internalization. We thus targeted the apically constricting zone. To do so we use IR fs laser dissection over a grid pattern that extends over the constricting zone (as presented in the Methods). This large ablation pattern disfavours actomyosin fast recovery that is otherwise unstoppable. We now mention this more specifically in the methods. As clarified previously, membrane signal reduction is caused by i) cells dilation driving membrane fluorophore density decrease and ii) bleaching which is more prominent for this specific experimental design because of the iterative protocol implemented. As indicated in the manuscript, the goal of these experiments is not to unveil something especially new, but to perform new and more direct experimental trials to corroborate the previously supported notion that apical constriction is necessary for VFF. We further emphasize this at the end of this section.

The authors next introduce slam dunk embryos that fail to cellularize but still make furrows. With no lateral and basal membranes/cellular structures, they argue this indicates that everything necessary for furrow formation must be driven by apical forces and build an elastic model for forces at the cell surface. They avoid the inclusion of viscosity in the model: they make the argument based on the observation that the relaxation time for the embryonic epithelium is 1 minute and furrow formation is considerably longer, and that the load generated by myosin is constantly increases.

Especially for readers who are primarily biologists, the rationale for this conclusion should be more fully developed, either with a few sentences, with references here, or more extensively in the supplement.

We now better developed the rationale of this in a new SI part, “Role of viscoelastic relaxation in the presence of an exponential increase in stress”.

The authors describe their continuum model for the surface of the embryo, and cite previous work, confirmed here on the distribution of apical myosin in the mesoderm. They next describe pre-strain that is proportional to myosin intensity. Unfortunately, explicit description of pre-strain is buried and deserves a paragraph of its own. It is not clear why pre-strain should be proportional to myosin intensity (or rather “normalized myosin excess fluorescence”) and that assertion should be developed more fully and explicitly. These details are important for full appreciation of the model.

We are sorry that Reviewer#2 could not find the description of pre-stress and pre-strain. They are included in “Box 1: definition of key terms”, which we hope will be visible to all readers. There is a paragraph in the main text describing mechanistically pre-strain in our context (pages 7-8, from L26) which refers to this help Box and to Supplementary Information for details.

The relation between MyoII and pre-stress/pre-strain is long established in the literature, see e.g. Chicurel et al, 1998, *Curr Op Cell Biol* doi:[10.1016/S0955-0674\(98\)80145-2](https://doi.org/10.1016/S0955-0674(98)80145-2), and has been discussed in the context of morphogenesis (e.g Inber 2006, *Int J Dev Biol*, doi: 10.1387/ijdb.052044di, Gjorevski and Nelson 2010, *Integ Biol* doi:10.1039/c0ib00040j). We have added additional references closer to our topic in the main text.

The authors next address their overall model, how tension anisotropy emerges from tissue and embryo geometry. Page 8, Line 11. It is not clear what “decomposing the length of the contracting ventral tissue” is. The authors need to explain this more explicitly to keep the reader on board. In this paragraph, the authors conclude that “the shape anisotropy of the system would thus explain why the surrounding tissue appears more difficult to deform along the AP than the DV direction even though the mechanical properties of the entire tissue surrounding the mesoderm are imposed to be the same.” This is a very interesting observation, and the next two paragraphs are designed to evaluate the consequences of these observations. They should be able to do this more explicitly.

We now reformulated this paragraph for clarity: “What is the origin of the stress anisotropy? To answer this question, we compared the dimension of the ventral tissue with respect to the dimensions of the entire blastoderm along the AP and DV axes. The ventral tissue is about three and six times less than the total blastoderm length along the mid-sagittal and mid-cross sections, respectively (Supplementary Fig.1 c).”

Page 8, Line 20. Do the authors really mean to say that “...actomyosin networks anchored to stiff boundaries which oppose resistance to deformation...”. Don’t they mean “...oppose deformation...”?

We now changed the expression to “... which resist deformation... ”.

Page 8, Line 27. The authors introduce “the trace of stress tensors” and “principle stresses, which are the eigenvalues of the stress tensor.” Again, the authors should present their findings in a more pedagogical fashion so that biologists can readily interpret their findings. This would require more extensive treatments of “traces of stress tensors”, principle stresses and eigenvalues of the stress tensor, probably in the supplement. For their model and findings to be most useful, they will be used by biologists to design new experimental tests that verify, modify or refute the model.

We share the Reviewer#2 position that making the paper explanatory for all audiences is important. We have now given further explanations in the SI that we hope will be helpful.

Fig. 3a makes a compelling case for the model providing a good fit for the biological data.

Page 10, Lines 8-10. The authors state “Remarkably, the 2D elastic sheet forms a buckle resulting in a furrow along the long axis of the 3D ellipsoid in the region under pre-stress (Fig. 4a and Supplementary Fig. 3a). This shows that forces applied at the surface of an ellipsoidal 3D shape can be sufficient to drive the formation of a furrow.” The authors should expand on this to state exactly where the forces that drive buckling are located, and what their nature is. Subsequent analysis of in toto tissue movements is compelling, but the authors leave the reader to speculate as to the nature of the forces that are driving buckling (or they believe they are so obvious as to not require explicit description).

We now put more effort to clarify this in the text. We would like to insist on the fact that it is the same active pre-stress in the model that gives rise to both the apical area changes depicted Fig 3 and the 3D shape changes, including furrow formation, depicted Fig 4-5.

Much later (page 12, line 12) the authors tell us that there is a contractile string that drives furrow formation. Are other forces involved? Is there a connection to cell autonomous forces in the cells that are in the furrow.

In our model no cell autonomous forces are involved (e.g., apical-basal differential tension or lateral tension). In the model, the pre-stress pattern, that reflects MyoII in vivo distributions (Fig. 1g) is imposed only at the apex of mesodermal cells.

As the pre-stress (proxy for MyoII activity) increases, tension builds up through mechanical balance, leading to *both* the cell areas and embryo shape change (Fig 3 and Fig 4 and 5, respectively).

We have now made this more explicit in the discussion, with the text p16 L3: ‘By imposing a ventral pre-stress proportional to MyoII distribution measured in vivo at the apical surface of cells, we show that our computational model can predict the magnitude and the dynamics both of furrow formation and of cell apical shape changes, which happen simultaneously in the model.’

The authors add biological test of this by cauterizing new anchors for the AP axis and show results consistent with their model. Ultimately, the authors do not address the relationship between furrow formation and invagination of the mesoderm and the convergent extension that begins to drive germ band extension. Are furrow formation,

tissue internalization completely separable events in time and space? If so, the authors need to state as much, perhaps in more than one place (e.g., in the Introduction and the Discussion). If not they need to tell us how their model for furrowing might be impacted by those other processes.

We now state this in the introduction at p2 L4 and in the discussion at p15 L23 as suggested by Reviewer#2. In the introduction we now write: “*VFF is eventually followed by ventral tissue internalization and germband extension. In vivo studies have highlighted several concurring phenomena during VFF.*” In the discussion we now write: “*Epithelial furrowing, eventually followed by tissue internalization, is a fundamental process during embryo gastrulation and neurulation.*”

Some additional, random (or reiterated notes)

The description for Fig 1 is inadequate

We now improved Fig.1 legend and corresponding methods.

Figure legend 2, the authors describe panel e before panel d.

This is now changed. We thank Reviewer#2 for these formatting remarks.

Reviewer#3 remarks to the authors

This manuscript is globally correct but rather incremental. It convincingly shows that buckling is sufficient to explain the observations. However, buckling is not by itself a new mechanism to explain folding.

We agree with the reviewer that other authors have demonstrated that other morphogenetic motions can be characterised as buckling. This is reviewed by Nelson (J Biomech Engng, 2016) who cites lung, intestine villi, tooth and brain morphogenesis. Note that in all these cases buckling occurs because of growth in confinement, which implies a very different sort of buckling compared to our case of tension-driven buckling of a curved surface.

In addition, as is well known, and as is shown in the authors' previous Fig 1 of Ref 19 (Rauzi Biophys J 2013), several models can explain the same phenotype.

We understand Reviewer#3 feedback but we do not agree on this point. Past studies have proposed deformations driven by apical-basal differential tension and/or lateral tension resulting in a bending moment (i.e., a torque force). These models rely on cells apical, basal and lateral side/tension and i) cannot explain why in mutants lacking lateral and basal cell sides a furrow can still form, cannot predict ii) the fact that the ventral furrow (VF) appears suddenly after MyoII signal has already significantly increased (threshold effect typical of buckling), iii) the AP flattening of the mesoderm tissue during VF formation and iv) that the VF propagates from medial to distal.

Our model can thus quantitatively predict key features of the VF that have been revealed in this study for the first time (e.g., ventral tissue AP flattening and furrow propagation). The model we propose is thus a major advance in the understanding of the mechanics and shaping of the VF.

In order to make this clear, we have now substantially rewritten the introduction.

Here, the current manuscript (despite its claim, e.g. in the abstract), does not convincingly show that buckling is really necessary to explain the phenotype. It rather shows that apical contraction is necessary to drive furrow formation.

We here agree with Reviewer#3: we do not show that embryo-scale buckling is necessary. We show that apical forces are necessary and that embryo-scale buckling, driven by apical forces, is sufficient to drive VF formation.

I am not sure that, after revision, it will be of interest for Nat Comm. I recommend to submit it to a more specialised journal.

Revisions :

Use consistent vocabulary. e.g. : laser dissection / ablation / manipulation (p2L20, Fig 1a, Fig 1c, p5L6, etc).

We thank Reviewer#3 for pointing this out. We now made the changes. At p2L20 we prefer using the expression laser manipulation since it includes both laser ablation and laser cauterization.

Each fig should be called in order (e.g.: P7L18, p7L21, p7L25), with a sentence which corresponds to the actual content of the figure (e.g.: p8L10, p9L27), and have an adequate title (eg should Fig 2 title be "Surface Evolver simulations" ? SuppFigs lack titles) ; make clear to the reader what is experimental and what is the result of simulations.

We now reordered the reference to figures and improved the clarity of the text.

Clarify (p12L8-9, p13L2, p15L24-27, p15L30-31, p16L5-6), remove redundancies (e.g.: p5L7=p5L10), correct typos ("Utricularia trap" missing p23L27).

We now clarified the text, removed redundancies and corrected typos as suggested by Reviewer#3.

The box should be cut in two boxes. A first box on stress and strain (currently part "b"). A second box on furrowing etc (currently part "a"). Both should be more legible.

We thank the reviewer for this suggestion. After carefully reviewing the options of making two Boxes out of one Box, we have decided to swap the order of parts a and b so that definitions and figure panels appear in the same order, but maintain the two parts together in the same Box.

Compare better with authors' previous papers, notably with Ref 19 (Rauzi Biophys J 2013) and Ref 5 (Dicko PLoSCB 2017).

We have expanded the text describing previous models, both with reference to our own work or other authors. Rauzi et al, 2013, is a review of VFF models. We have improved the text where we cite it, and we explicitly refer to it for explanations of the common point of those models and for results anterior to 2013. Dicko et al, 2017, focus on a later stage of Drosophila morphogenesis where motion is tangential to the embryo surface, which we do not think we can discuss within the scope of this paper, however they have in common with our present work that they take into account the whole embryo (as Streichan et al 2018 and Saadoui 2020 do).

Also, note the preprint : bioRxiv preprint doi:<https://doi.org/10.1101/2021.09.16.460711>

We thank Reviewer#3 for pointing out this paper in BiorXiv that we now reference in the discussion. "New imaging technology provides a synthetic view of the coordination of tissues at the scale of the whole embryo with subcellular resolution [1, 4, 3, 2, Stern et al. Biorxiv 2021]."

POLICIES AND FORMS REQUIRED FOR RESUBMISSION

** Please complete or update the following checklist(s) to verify compliance with our research ethics and data reporting standards. Address all points on the checklist, revising your manuscript in response to the points if needed.*

The form(s) must be downloaded and completed in Adobe Reader rather than opened in a web browser. Each form must be uploaded as a Related Manuscript file at the time of resubmission.

Editorial policy checklist:

<https://www.nature.com/documents/nr-editorial-policy-checklist.pdf>

Reporting summary:

Files uploaded.

** Your paper uses custom code/software. Please complete the following code and software submission checklist and make your code available for reviewer assessment, if you have not already done so. The code/software can be provided in a zip file with a readme.txt file or other instructions for installing and running the software. If appropriate, also provide example data and expected output. If you have any issues with the file upload, please let me know.*

<https://www.nature.com/documents/nr-software-policy.pdf>

File uploaded. Note that the software and its manual are all available at <https://zenodo.org/record/5911338> and <https://zenodo.org/record/6043524> and have thus not been uploaded.

DATA AND CODE AVAILABILITY

** All Nature Communications manuscripts must include a “Data Availability” section after the Methods section but before the References. If any of the data can only be shared on request or are subject to restrictions, please specify the reasons and explain how, when, and by whom the data can be accessed. For more information on this policy and a list of examples, see:*

<https://www.nature.com/documents/nr-data-availability-statements-data-citations.pdf>

** Please also include a “Code Availability” section after the “Data Availability” section. If the code can only be shared on request, please specify the reasons. For more information on our code sharing policy and requirements, please see:*

** We strongly encourage you to deposit all new data associated with the paper in a persistent repository where they can be freely and enduringly accessed. We recommend submitting the data to discipline-specific and community-recognized repositories; a list of repositories is provided here: <http://www.nature.com/sdata/policies/repositories>*

** To maximise the reproducibility of research data, we strongly encourage you to provide a file containing the raw data underlying the following types of display items:*

- Any reported means/averages in box plots, bar charts, and tables*
- Dot plots/scatter plots, especially when there are overlapping points*
- Line graphs*

The data should be provided in a single Excel file with data for each figure/table in a separate sheet, or in multiple labelled files within a zipped folder. Name this file or folder ‘Source Data’, and include a brief description in your cover letter. The “Data Availability” section should also include the statement “Source data are provided with this paper.”

To learn more about our motivation behind this policy, please see: <https://www.nature.com/articles/s41467-018-06012-8>

** Please replace your bar graphs with plots that feature information about the distribution of the underlying data. All data points should be shown for plots with a sample size less than 10. For larger sample sizes, please consider box-and-whisker or violin plots as alternatives. Measures of centrality, dispersion and/or error bars should be plotted and described in the figure legend.*

ORCID

** Nature Communications is committed to improving transparency in authorship. As part of our efforts in this direction, we are now requesting that all authors identified as ‘corresponding author’ create and link their Open Researcher and Contributor Identifier (ORCID) with their account on the Manuscript Tracking*

System prior to acceptance. ORCID helps the scientific community achieve unambiguous attribution of all scholarly contributions.

You can create and link your ORCID from the home page of the Manuscript Tracking System by clicking on 'Modify my Springer Nature account' and following these instructions. Please also inform all co-authors that they can add their ORCIDs to their accounts and that they must do so prior to acceptance.

If you experience problems in linking your ORCID, please contact the Platform Support Helpdesk.

REVIEWERS' COMMENTS

Reviewer #1 (Remarks to the Author):

The authors have addressed some of my comments. This is a re-write of the original manuscript, with no new experiments except for the addition of Figure S3g. I appreciate the authors' effort to reference and discuss some of the previous work in this revision. I think the differential contribution of the model (they can explain folding in acellular embryos) is better spelled out now. My main caveat remains that the biological significance of the findings in the manuscript is incremental.

Reviewer #3 (Remarks to the Author):

The authors have carefully revised the manuscript. It is now fluid, clearer (except maybe for Box 1) and does justice to their work, including to the predictive power of the model. I recommend to publish it.

NB :

- Main Figs have captions in the color-marked manuscript but not in the black&white one.
- Supp Figs still lack titles.

Reviewer #1 (Remarks to the Author):

The authors have addressed some of my comments. This is a re-write of the original manuscript, with no new experiments except for the addition of Figure S3g. I appreciate the authors' effort to reference and discuss some of the previous work in this revision. I think the differential contribution of the model (they can explain folding in acellular embryos) is better spelled out now. My main caveat remains that the biological significance of the findings in the manuscript is incremental.

We thank the reviewer for his reading.

Reviewer #3 (Remarks to the Author):

The authors have carefully revised the manuscript. It is now fluid, clearer (except maybe for Box 1) and does justice to their work, including to the predictive power of the model. I recommend to publish it.

We thank the reviewer for his reading.

NB :

- Main Figs have captions in the color-marked manuscript but not in the black&white one.
- Supp Figs still lack titles.

This has been corrected.